

# Geophysical and geochemical controls on the megafaunal community of a high Arctic cold seep

Arunima Sen[1], Emmelie K.L. Åström[1], Wei-Li Hong[1,2], Alexey Portnov[1,3], Malin Waage[1], Pavel Serov[1], Michael L. Carroll[1,4] and JoLynn Carroll[4]

[1]Centre for Arctic Gas Hydrate, Environment and Climate (CAGE), Department of Geosciences, UiT The Arctic University of Norway, Tromsø, 9037, Norway

[2]Geological Survey of Norway (NGU), Trondheim, 7491, Norway

[3]School of Earth Sciences, Ohio State University, Columbus, Ohio, 43210, U.S.A.

[4]Akvaplan-niva, FRAM - High North Research Centre for Climate and the Environment, Tromsø, 9296, Norway

*Correspondence to*: Arunima Sen (arunima.sen@uit.no)

**Abstract**

Cold seep communities around gas hydrate mounds (pingos) in the Western Barents Sea (76°N, 16°E, ~400 m depth) were investigated with high resolution, geographically referenced images acquired with an ROV and towed camera. Four pingos associated with seabed methane release hosted diverse biological communities of mainly non-seep (background) species including commercially important fish and crustaceans, as well as a species new to this area (the snow crab *Chionoecetes opilio*). We attribute the presence of most benthic community members to habitat heterogeneity and the occurrence of hard substrates (methane derived authigenic carbonates), particularly the most abundant phyla (Cnidaria and Porifera), though food availability and exposure to a diverse microbial community is also important for certain taxa. Only one chemosynthesis based species was confirmed, the siboglinid frenulate polychaete, *Oligobrachia haakonmosbiensis*. Overall, the pingo communities formed two distinct clusters, distinguished by the presence or absence of frenulates. Methane gas advection through sediments was absent below the single pingo that lacked frenulates, while seismic profiles indicated gas saturated sediment below the other frenulate colonized pingos. The absence of frenulates could not be explained by sediment sulfide concentrations, despite these worms likely containing sulfide oxidizing symbionts. We propose that high levels of seafloor methane seepage linked to sub-surface gas reservoirs support an abundant and active sediment methanotrophic community that maintains high sulfide fluxes and serves as a carbon source for frenulate worms. The pingo currently lacking a sub-surface gas source and lower methane concentrations has lower sulfide flux rates and limited amounts of carbon insufficient to support frenulates. Two previously undocumented behaviors were visible through the images: grazing activity of snow crabs on bacterial mats, and seafloor crawling of *Nothria conchylega* onuphid polychaetes.



# 1 Introduction

Cold seeps, where hydrocarbons and reduced gases emerge from the seafloor, are ubiquitous in the world's oceans and despite being discovered only a few decades ago (Paull et al., 1984), they have been studied intensively in a variety of settings around the world (Levin, 2005; Levin et al., 2016; Sibuet and Olu, 1998; Sibuet and Olu-Le Roy, 2002). However, cold seeps in the

Arctic Ocean have received less attention and the literature on Arctic seep communities is limited to a few studies in the Barents and Beaufort Seas (Åström et al., 2016, 2017b, 2017a; Gebruk et al., 2003; Lösekann et al., 2008; Paull et al., 2015; Pimenov et al., 2000; Rybakova (Goroslavskaya) et al., 2013). The most well studied seep site in the Arctic is the Håkon Mosby mud volcano (HMMV), which has practically become synonymous with Arctic seep biology. Paradoxically, high thermal gradients in the sediment have led researchers to conclude that HMMV does not really constitute a typical cold seep

(Gebruk et al., 2003).

Another limitation to our current understanding of cold seeps is the focus on mainly deep-sea sites. It should be noted that the terms 'shallow' and 'deep' are relative, and a strict, universally accepted cutoff value separating the two does not exist. Nonetheless, relatively shallow seeps, such as those on continental shelves and upper continental slopes, have not been studied nearly as well as their deep-sea counterparts. In their reviews of cold seeps, Sibuet and Olu (1998; 2002) only considered sites

at a minimum of 400 m water depth and even the more recent review of Levin et al. (2016) refers to cold seeps within the context of the deep sea. Yet studies of seeps in comparatively shallow water (< 400 meters) are crucial to resolve depth-related trends in biodiversity, chemosymbiotic species and seep-obligate fauna (Carney et al., 1983; Dando, 2010; Sahling et al., 2003).

Several sites of methane seepage have been discovered on the continental shelf offshore Svalbard and in the northwest Barents

Sea (Andreassen et al., 2017; Åström et al., 2016; Portnov et al., 2016; Sahling et al., 2014; Serov et al., 2017). An abundance of cold seeps in the Arctic is important, because the Arctic is connected to both the Pacific and the Atlantic Oceans. This setting provides an excellent opportunity to study the establishment of biogeographic provinces, migration and connectivity between seep populations that are otherwise disconnected from each other at lower latitudes. The presence of numerous cold seeps on the Barents Sea shelf could also be pertinent to the overall ecology and economy of the Arctic. The Barents Sea is

considered an ecological hotspot for the circumpolar Arctic and an economically important region supporting one of the richest fisheries in the world (Carroll et al., 2018; Haug et al., 2017; Wassmann et al., 2011). The interaction between cold Arctic and warm Atlantic water masses, seasonal sea ice cover and the interplay of pelagic-benthic coupling creates a highly productive region (Degen et al., 2016; Ingvaldsen and Loeng, 2009; Sakshaug et al., 2009; Tamelander et al., 2006). Moreover, the Arctic and particularly the Barents Sea, are predicted to experience amplified impacts of climate warming such as shrinking sea ice

cover, changing oceanographic patterns and increasing ocean acidification (Haug et al., 2017; Onarheim and Årthun, 2017; Węsławski et al., 2011). Such climatic and environmental changes in the region and the associated impact of newly established invasive and northward migratory species may cause major ecological shifts in the Barents Sea (Cochrane et al., 2009; Degen





et al., 2016; Johannesen et al., 2012). With our limited knowledge of the biology and ecology of Arctic seeps, predictions about how these methane based ecosystems will respond to a warming Arctic are difficult to make.

This study examines the megafaunal community associated with a cold seep site on the Arctic shelf in the western Barents Sea (Fig. 1). Our results serve as a first step towards addressing some of the existing gaps of knowledge regarding cold seep and Arctic ecology, i.e., with respect to seeps on the continental shelf in relatively shallow water (<400 m) in the high Arctic (76ºN) (Åström et al., 2016, 2017b; Dando, 2010; Paull et al., 2015).

High resolution, georeferenced seabed imagery was used for analyzing the communities associated with four gas hydrate bearing mounds (pingos) exhibiting active methane seepage. All animals visible in the images (i.e., at least a few cm in size) were examined, thereby resulting in the inclusion of different categories of animals such as epifauna, infauna and even some pelagic species. Hydroacoustic surveys revealed flares of gas rising into the water column from around the summits of three of the four investigated pingos, suggesting different seepage regimes and sediment geochemical conditions between the free gas emitting pingos and the single pingo from which no flares were seen. We hypothesized that megafaunal communities at free gas emitting and non-emitting pingos would differ. Further, we expected differences in the composition of sulfide and methane in sediment pore water between the gas emitting pingos and non-emitting pingo, to account for differences in associated megafaunal communities. The setting for this study is particularly useful for teasing apart the factors affecting the distribution of chemosynthesis based species, since these animals are directly reliant on seeping chemicals (Levin, 2005; Sibuet and Olu, 1998). Chemosynthesis based animals are often considered ecosystem engineers within cold seep systems and their presence or absence may subsequently affect community structure as a whole (Cordes et al., 2010; Levin, 2005; Levin et al., 2016). Our approach consisted of linking seepage patterns to sediment geochemistry and the distribution patterns of chemosynthesis based animals in the context of the overall community structure.

## 2 Methods and Materials

### 2.1 Study site

The area of focus for this study is a site on the Arctic shelf (hereafter referred to as the 'pingo site'), about 50 km south of Sørkapp (South Cape), Spitsbergen, characterized by sub-circular, domed seabed structures (Fig. 1) from which gas hydrates have been recovered in sediment cores (Hong et al., 2017). The morphological similarity of these mounds to terrestrial and offshore pingos have resulted in them being referred to as 'gas hydrate pingos' (GHPs) (Serov et al., 2017). Originally, the term pingo refers to mounds of earth-covered ice in permafrost regions, formed by the hydrostatic pressure of water in the permafrost (Pissart, 1985). Similar features in marine systems, where sediment gas hydrates are analogous to ice in terrestrial systems have been referred to as gas hydrate pingos or submarine pingos (Chapman et al., 2004; Hovland and Svensen, 2006;



Paull et al., 2007; Serié et al., 2012). In this study, the term gas hydrate pingos (GHPs), or simply, pingos, will be used for the 4 features of interest

The pingo site is located at a depth of about 380 m, on the flank of the glacially eroded Storfjordrenna cross shelf trough. A stable grounded ice sheet over Storfjordrenna, followed by alternating warm and cold periods resulted in both the accumulation

of gas hydrates as well as their episodic dissociation over the past 22,000 years (Serov et al., 2017). The GHPs themselves are proposed to have been formed ~15,500 years ago, when deglaciation followed by a warm Heinrich H1 event had a particularly debilitating effect on the gas hydrate stability zone (GHSZ) and resulted in the large scale release of methane from gas hydrates that had accumulated during the prior thousands of years. Since about 8,000 years ago, however, the region experienced a steady transition to current conditions of stable gas hydrates (Serov et al., 2017).

This study focuses on a cluster of four GHPs, within an area of 2 km². These GHPs rise gradually above the surrounding seafloor (8-12 m) with diameters ranging between 280 m and 450 m. Hydroacoustic, seismic and geochemical surveys show persistent and continuous release of predominantly thermogenic methane gas around the summits of three of the four GHPs (GHPs 1,2 and 3) (Serov et al., 2017). No such free gas emissions were seen over GHP5 during repeat on-site observations over 3 years (2013-2016) during different seasons.

**2.2 Imagery**

Two sets of seafloor imagery were collected in 2015 and 2016 (Fig. 1). The first set was taken in 2015, with the MISO-WHOI (Multidisciplinary Instrumentation in Support of Oceanography, Woods Hole Oceanographic Institution) towed camera (tow cam for short) aboard the R/V *Helmer Hanssen* (cruise number CAGE-15_2). The tow cam consisted of a 16 Megapixel digital still camera with optical image stabilization (photo resolution: 4928 x 3264 pixels). It was mounted on a frame that also

contained 6 cores (~ 1 m long) and 6 Niskin bottles. Due to space limitations and the logistical difficulties with mounting the cores and the camera together on the main body of the frame, the downward facing camera was tilted by 25 degrees. Images were taken every 10-15 seconds. Despite slow ship speeds, overlap between successive images could not be achieved, therefore the tow cam image surveys were essentially transects over the different GHPs. The dataset consisted of one transect each over GHP1 and GHP2, and two transects over GHP3. Transects were named with an acronym for tow cam, followed by the dive

number and the pingo number (e.g., TC25 GHP1). Navigation files from transects over GHP3 were inadequate for georeferencing purposes, therefore the images associated with these transects were only used qualitatively to ascertain species' presence or absence.

The second set of images was taken in 2016, also aboard *Helmer Hanssen* (cruise number CAGE-16_5). During this cruise,

images were acquired via a pair of stereo cameras mounted on a remotely operated vehicle (ROV), 30K, operated by the Norwegian University for Science and Technology (NTNU). The stereo cameras (GC1380 digital still cameras, image



resolutions of 1360 x 1024 pixels) were spaced 40 cm apart linearly, ensuring more than 50% overlap between left and right cameras, and faced downward at an angle of 35 degrees. Due to higher maneuverability and control over an ROV in comparison to the tow cam, the imagery surveys in 2016 were conducted with the purpose of constructing mosaics (i.e., overlapping images taken in a lawn mower fashion). Three mosaicking surveys were conducted over GHP5. Mosaics were named ROV, followed

by the mosaic number and GHP5 (e.g., ROV1 GHP5). Navigation at GHP3 was unreliable, therefore the corresponding images were unusable for quantitative analyses. However, these images were used to conduct a comparison of animals visually identifiable in the tow cam and ROV images.

## 2.3 Mosaicking and georeferencing

Neither the tow cam system nor the ship had closed loop positioning systems during the 2015 cruise. Using the length of the

tow cam system's cable to correct image location proved unsuccessful, therefore the ship's coordinates were used for positioning the tow cam images in space. At the scale of the site, this level of georeferencing is more than adequate, for it could be used to differentiate between different pingos and overall locations over them (summits, flanks, etc.). The ROV images were georeferenced based on coordinates obtained through an ultra-short baseline (USBL) closed positioning system. Images were mosaicked with the IFREMER software, Matisse v3 (courtesy Aurélien Arnaubec). This software takes angles

of tilt into account for estimating the footprints of images on the seafloor and uses navigation data for placing the mosaics in space. In the case of the tow cam images, since no overlap existed between images, the GeoTIFF output from Matisse v3 consisted of single images in space based on the coordinates of the image (Fig. 2). With the ROV images, the software produced a georeferenced mosaic as the GeoTIFF output. Due to the low quality blending process of Matisse, higher quality seamless mosaics using a customized mosaicking script within Matlab (Pizarro and Singh, 2003; Singh et al., 2004) which were

subsequently georeferenced by matching and lining up easily identifiable features to the same features in the Matisse mosaics (Fig. 2). All georeferenced images and mosaics were displayed within ArcGIS (ArcMap 10.3 and 10.5)

## 2.4 Faunal identification and community analyses

Visible fauna (at least a few centimeters across) were identified to the lowest possible taxonomic division and marked manually (Table 1). Different categories of fauna were included, and therefore the biological communities examined in this study are

referred to as megafauna and by that, we mean animals large enough to be seen with the naked eye (Danovaro, 2009). While the majority of fauna would be considered epifauna, animals partially buried in the sediment were also included and a few species were present on the seafloor that could more generally be considered pelagic (e.g., ctenophores). Each individual was marked and the numbers were standardized to the different areas of the mosaics and transects by converting to densities based on the size of the mosaic or transect area.

Numerous individuals of siboglinid worms were seen and since specimens collected in core samples were found to be frenulates lacking pinnules on the tentacles, the species identity was narrowed down to two possibilities: *Oligobrachia*

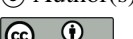



*haakonmosbiensis* or *Oligobrachia webbi* (Brattegard, 1966; Smirnov, 2000). These two species are similar in terms of morphology and while officially they are considered different species (Smirnov, 2000, 2014), a consensus does not exist on this separation (e.g., Meunier et al., 2010). Since no DNA sequences are available for *O. webbi*, similar worms from other Arctic seeps have been referred to as *O. haakonmosbiensis* due to close affinities in mitochondrial COI sequences with this

species (Lösekann et al., 2008; Paull et al., 2015). For the sake of consistency with other Arctic seep studies, we will refer to the species in this study as *O. haakonmosbiensis*. It was not feasible to count individual frenulates in the mosaics or image transects. Therefore, aggregations were outlined and the average density of 7000 individuals/m$^2$ obtained from core samples taken during in 2016 was applied to the aggregation outlines in order to estimate numbers of individuals within aggregations and densities in the transects or mosaics as a whole.

Similarly, every individual of the small solitary *Caryophilia* coral could not be marked in the images. Aggregations of the two morphotypes (pink and white) of this taxon were outlined, similar to the siboglinids, however, densities from samples could not be applied to the outlined aggregations for estimating numbers of individuals since *Caryophilia* tends to populate hard surfaces which were avoided during core sampling. Instead, six *Caryophilia* aggregations were selected at random for each morphotype from the mosaics and image transects, and the number of individuals present in each aggregation were counted.

Among the randomly chosen aggregations, on average, 27 individuals of the pink morphotype were seen (within an average aggregation size of 116 cm$^2$) and 28 individuals of the white morphotype were seen (within an average aggregation size of 34 cm$^2$). These averages were used to estimate total numbers and overall densities for all remaining aggregations outlined in the mosaics and image transects.

An exception to the standard methodology of marking every visible taxon, was a type of encrusting animal, possibly a
bryozoan. Reddish, brownish and greenish morphotypes of this animal were seen, but given the difficulty in identifying them or even visualizing them sufficiently, these animals were not marked in the image transects and mosaics nor discussed in this study.

The tow cam images captured more detail than the ROV stereo cameras. Therefore, before inclusion in the community analysis, the different taxa were evaluated both in terms of size and shape as well as their overall ability to be seen in images collected
by the ROV stereo cameras. Twenty taxa identified in tow cam images were not seen in the ROV images. Of these taxa, twelve were rare, i.e. only seen once, or at one site. These rare taxa were included in the community analyses because their absence from the ROV mosaics is likely not due to their inability to have been seen on the images, but rather due to their rare appearance. In order to determine whether the remaining 8 should be included in the community analyses, the ROV images from pingo 3 (with unreliable navigation data) were examined. Comparing the ROV GHP3 and ROV GHP5 images made it
possible to explore whether animals not seen in the ROV GHP5 images was due to an inherent inability to see them in ROV images because of their lower resolution. Of these, juvenile fish, *Molpadia borealis* sea cucumbers, white possible scaphopods, the two unidentifiable polychaete species and the onuphid worm, *Nothria conchylega* were not seen in the ROV GHP3 images. *M. borealis* sea cucumbers and juvenile fish tend to stick out more, and are larger than some of the smallest animals seen in





the ROV images such as stick sponges and *P. borealis* shrimp. Therefore they were considered detectable through the ROV stereo cameras and their absence from an ROV image was attributed to their actual absence and not due to their inability to be seen in those images. They were therefore included in the community analyses. The other animals that were not seen in the ROV GHP3 images were excluded from the community analyses because they likely would have been missed despite being

present either because of their small size (e.g. white possible scaphopod) or because they blended into the background sediment (e.g. *N. conchylega*). In a few cases, this comparison between ROV images from GHP5 and GHP3 determined whether higher level taxa should be used. For example, pycnogonids were grouped together into a single category for community analyses despite at least three different morphotypes being visible in the tow cam images. Gastropods were all grouped together despite 10 morphotypes being visible in the tow cam images. Furthermore, hermit crabs (*Pagarus* sp.) were included in the gastropod

category since it was not always clear whether gastropod shells were occupied by the original inhabitants or by hermit crabs. Similarly, all ophiuroids were grouped together, as were some of the zoarcid fish.

Overall densities of visible fauna were used in creating a Bray Curtis similarity matrix, which was the basis for multi-dimensional scaling (MDS) and cluster analyses. A fourth root transformation was applied on the abundance data due to the

vast range of densities among faunal groups, in order to balance the impact of both highly abundant and rare taxa within the same dataset. MDS/cluster analysis was conducted both with and without the inclusion of frenulates. In both iterations of community analyses ANOSIM tests were run in order to test the significance of the cluster results.

**2.5 Geochemical measurements**

Gravity cores were recovered from the different GHPs in order to determine the geochemical characteristics of sediment pore

fluids (Fig. 1 and Table 2). In 2015, six gravity cores were taken: cores 911, 912 and 940 at GHP3, core 913 at GHP1, core 914 at GHP2 and core 920 at GHP5. Sulfide, sulfate, dissolved inorganic carbon (DIC) and methane were measured in cores 911 and 920. Methane was not measured in core 940, and only methane was measured in cores 912, 913 and 914. In 2016, 5 gravity cores were taken at GHP3 and GHP5. Core 1045 at GHP3 and cores 1048, 1068, 1069 and 1070 at GHP5. All 2016 cores were subjected to the full array of geochemical analyses with the exception of core 1048, for which methane was not

measured.

**2.5.1 Porewater measurements**

Details of the porewater sampling and analyses are given in Hong et al. (2017). Briefly, porewater samples were collected by inserting acid-washed rhizons into pre-defined depths of the sediment cores in the refrigerated room onboard. Quantitites of 5-15 ml of water were collected in acid-washed syringes. The total alkalinity was measured by Gram titration method within

two hours after the syringes were disconnected from the rhizons. An aliquot of water sample was preserved with saturated $Zn(OAc)_2$ solution to prevent re-dissolution of sulfde within half an hour. Samples were stored frozen until analyses were





conducted in the lab. Concentrations of hydrogen sulfide were analyzed spectrophotometrically using the 'Cline method' (Cline, 1969). Sulfate concentrations were measured from the same samples for sulphide analyses by ion chromatography. Concentrations of dissolved calcium and magnesium were measured by ICP-AES. Both analyses were performed in the laboratory of the Geological Survey of Norway (NGU).

**2.5.2 Inorganic carbon measurements**

The concentration of DIC was approximated based on carbonate alkalinity which itself was estimated by subtracting total alkalinity from the concentration of hydrogen sulfide. This is a reasonable assumption for the slightly basic porewater as bicarbonate ions should be dominated by carbonate species. To differentiate the different pathways of sulfate reduction, either through organic matter degradation or through coupling with anaerobic oxidation of methane (AOM), we estimated the values

of $\Delta$DIC and $\Delta$SO$_4$ based on the following equations (Claypool et al., 2006; Suess and Whiticar, 1989):

$$\Delta DIC = ([DIC]_{spl} - [DIC]_{BW}) + \Delta Ca^{2+} + \Delta Mg^{2+}$$

$$\Delta SO_4 = [SO_4]_{spl} - [SO_4]_{BW}$$

$$\Delta Ca^{2+} = [Ca^{2+}]_{spl} - [Ca^{2+}]_{BW}$$

$$\Delta Mg^{2+} = [Mg^{2+}]_{spl} - [Mg^{2+}]_{BW}$$

where $[]_{spl}$ and $[]_{BW}$ are the concentrations of various chemical species in porewater samples (spl) and bottom water (bw), respectively. We applied these calculations only to samples above the depth of the sulfate methane transition zone (SMTZ). Such operation assumes that DIC is produced by organoclastic sulfate reduction and/or AOM-sustained sulfate reduction while

carbonate precipitation, which consumes both calcium and magnesium, decreases DIC concentration. By plotting $\Delta$DIC against $\Delta$SO$_4$, it is possible to differentiate the primary DIC production reactions based on different stoichiometric relationships. For every mole of organic matter degraded, one mole of sulfate is consumed and two moles of DIC will be produced. While for AOM-sustained sulphate reduction, one mole of DIC is produced for every mole of sulfate consumed.

**2.5.3 Methane measurements**

Concentrations of dissolved pore water methane were obtained through conventional headspace sample preparation (Kolb and Ettre, 2006) and flame-ionized detector gas chromatography. The bulk sediment volume of 5ml was placed in 20 ml headspace vials, 5 ml of 1 molar NaOH solution was added, the vials were capped with rubber septa, sealed with aluminum crimp caps and shaken for 2 minutes. The measurements were carried out using TG-BOND Alumina (Na$_2$SO$_4$) 30m x

0.53mm x 10µ column on ThermoScientific Trace 1310 gas chromatograph.





## 2.6 Seismic data

A seismic profile was generated from a broadband (10-350 Hz) high-resolution (~6 m lateral- and 2-3 m vertical resolution of the shallow subsurface) P-Cable 3D seismic cube (R/V *Helmer Hanssen* cruise number CAGE 16-6, 2016). This P-Cable 3D seismic system consisted of fourteen, 25 m long, streamers each containing 8 hydrophones. The streamers were spaced 12.5 m apart along a cross-cable towed perpendicular to the ships steaming direction (Petersen et al. (2010) and Waage et al. in prep.).

## 3 Results

### 3.1 Community characteristics

Bacterial mats were seen in every image transect or mosaic, confirming the presence of reduced chemicals in the sediment and seepage activity at every pingo including GHP5. Fewer bacterial mats were seen on GHP5 in comparison to the other GHPs (Table 1). Hard substrates were also seen in every image transect or mosaic. Most of these hard substrates were clearly authigenic carbonates, distinguishable by their pitted texture. The texture of all rock like features was not always visible in the images, either due to resolution issues or because of animals colonizing the surfaces. Therefore, some of the hard features could possibly be dropstones or other substrates not formed as a result of seepage activity. Nonetheless every GHP hosted carbonate structures, indicative of long term gas seepage.

A total of 60 taxa were identified and marked in the images (Table 1). Of these, 56 were used in the community analyses (see Section 2.4). On average, 29 taxa were seen in single mosaics or image transects over the different pingos (this average is based on raw richness counts and does not account for the differences in areas covered by the mosaics and image transects). Many morphologically distinct taxa were grouped together as a single taxon on a number of occasions. For example, at least 10 morphologically distinct types of gastropods and three types of pycnogonids were seen. Therefore, the total taxnonomic richness of the pingo site is likely considerably higher than the 60 taxa listed in Table 1.  Richness counts were similar between the various transects or mosaics, and furthermore, richness counts of the mosaics from the non gas emitting pingo (GHP5) were very similar to those at the gas emitting pingos (Table 3). On the other hand, the diversity indices revealed that the gas emitting pingos (except the TC25 transect over GHP3) were much less even than GHP5. This trend was only observed with siboglinids included in the analysis, with their removal, diversity indices did not display much variation between the individual pingos, mosaics or image transects. In addition to the 60 taxa seen and marked in the mosaics/transects, two individuals of *Chionoecetes opilio* (snow crab) and a few individuals of the wolffish (*Anarhichas minor*) were seen in the images over GHP3, (Table 1).



In total, 11 animal phyla were seen (Porifera, Cnidaria, Ctenophora, Nemertea, Priapulida, Sipuncula, Annelida, Arthropoda, Mollusca, Echinodermata, and Chordata). Of these, Cnidarians were represented by the largest number of taxa (18), followed by Echinoderms (11 taxa) and Chordates (10). The most abundant taxon by far was the frenulates, despite their absence from GHP5. The frenulate worms are the only known chemosynthetic species observed in this study and also the only potential seep

specific or obligate species. Following frenulates, the next most numerous taxa were *Gersemia* corals and *Thenea* sponges (likely *Thenea valdiviae,* Cárdenas and Rapp, 2012; Steenstrup and Tendal, 1982). The onuphid worm, *Nothria conchylega* was also abundant in the tow cam images and trails in the sediment were also seen behind these worms. Most of the cnidarians and the sponges were seen on hard substrates. A number of other animals were also seen on or near hard substrates, such as the Northern shrimp *Pandalus borealis*, ophiuroids and various gastropods. Pycnogonids were often seen among frenulate

worm tufts. Conversely, pycnogonids were rare or absent in image transects or mosaics where frenulates were not present. Among the various taxa, several hold economic value, such as Atlantic cod (*Gadus morhua*), Northern shrimp, haddock (*Melanogrammus aeglefinus)* and Greenland halibut (*Reinhardtius hippoglossoides*) (Norwegian Directorate of Fisheries, 2017).

The community analysis including frenulates resulted in communities on mosaics/transects separating into two distinct clusters (R = 0.926, p = 0.03, Fig. 3A). This clustering corresponded to communities containing the frenulates and communities without frenulates. In other words, the GHP5 mosaics, and the TC25 transect over GHP3 formed one cluster, while the other GHP3 transect and the transects over GHP1 and GHP2 formed a separate cluster. When frenulates were excluded from the analysis, a similar result was obtained, with two clusters corresponding to communities/mosaics from GHP5 and communities/transects

from the other GHPs. This clustering was also significant (R = 0.704, p = 0.03, Fig. 3B).

**3.2 Sediment geochemistry and sub-surface features**

Sulfide was below the detection limit in the bottom water as well as in the upper few centimeters of the sediment from all analyzed cores. It was first detected between depths of 5 and 10 cm, depending on the rate of Fe (II) production through iron reduction in the top layer of sediments. Subsequently, sulfide concentrations rapidly increased downcore, with high milimolar

level concentrations measured across all cores (Fig. 4A). The sulfide profiles of cores taken from GHP5 did not appear to differ substantially from sulfide profiles of cores from GHP3 (Fig. 4A). Methane was detectable at the sediment surface and often remained more or less constant, until large increases were measured below 40 cm (50 cm for GHP5 cores). At depth, methane concentrations tended to be lower in cores taken from GHP5 in comparison to the cores from the other pingos (Fig. 4B).

At both GHP3 and GHP5, by and large, a 1:1 correspondence was observed between increase in inorganic carbon and increase in sulfate in both shallow and deeper sediment, suggesting that most sulfate reduction in the sediment, is linked to anaerobic




oxidation of methane (AOM). In some cores, a switch from a 2:1 to a 1:1 ratio was observed (e.g., core 940, Fig. 5). Core 1045 from GHP5 was the only core in which even values from deeper in the sediment corresponded more closely to the 2:1 ratio, indicative of organic matter degradation via sulfate reducing bacteria being the major consumer of sulfate. Therefore, at both GHP3 and GHP5, sulfate reduction is coupled to methane oxidation, though in the shallow sediment, sulfate can also be

consumed by though the use of organic matter (Fig. 5). Further, the rate of sulfate consumption appears to differ between GHP3 and GHP5. Sulfate flux rates ranged from -0.31 mol/m$^2$/day to -2.08 mol/ m$^2$/day in cores from GHP3. The lowest rate of sulfate flux was measured in a core from GHP5 (-0.12 mol/m$^2$/day) and the maximum rate of sulfate flux measured in cores from GHP5 was only -0.9 mol/m$^2$/day. On average, the rate of sulfate flux measured in cores from GHP5 (-0.57 mol/m$^2$/day) was lower in comparison to cores from GHP3 (-1.22 mol /m$^2$/day, Table 4).

Beneath the three pingos emitting gas into the overlying water column, the seismic data revealed vertical zones of acoustic blanking in the shallow subsurface (up to ~150 m depth) and adjacent local high amplitude anomalies. In contrast, no anomalies or zones of acoustic blanking were observed underneath GHP5 (Fig. 6).

## 4 Discussion

Four gas hydrate pingo features within a 2 km$^2$ area on the Arctic shelf were examined for this study. Split beam echosounder data revealed gas bubbles in the water column above three of the four GHPs, often reaching impressive heights of a hundred meters above the seafloor (Serov et al., 2017). The emitted gases are primarily methane of thermogenic origin, confirming that these GHPs (GHP1, GHP2, GHP3) represent highly active methane seep sites. Although gas emissions into the water column were not detected at GHP5, the presence of bacterial mats indicates that this pingo also exhibits seepage. Carbonate formations,

including large slabs, were seen on all four pingos, suggestive of long term seepage (Berndt et al., 2014; Crémière et al., 2016; Seabrook et al., 2017). In this study, key attributes of these shallow water (<400 m) Arctic cold seep megafaunal communities were revealed and the distribution of chemosynthesis based community members was linked to seepage patterns and sub-surface features.  Comparisons of features of shallow water Arctic cold seeps identified through the present study with other seep sites indicate major differences between cold seep communities at high and low latitudes. We conclude by considering

cold seeps within the larger perspective of the Arctic, both from an ecological and economic point of view.

### 4.1 Community characteristics

Sixty-two megafaunal taxa were identified in total at the pingo site and on average, 29 taxa were seen within a single mosaic or image transect. No major differences in taxonomic richness between individual pingos was seen, though the image transects

containing siboglinid frenulates were much less even in terms of species abundances compared to the transect and mosaics





which did not contain any frenulates (Table 3). This is clearly due to the extremely high abundance of frenulates (thousands of individuals), and with this group excluded, diversity indices of the various pingos are comparable. It should be noted that species richness, and even morphospecies richness counts are considerably higher because in certain cases, morphospecies were grouped together under a single category. In one instance, this larger level grouping even lumped two different phyla

together (gastropod molluscs with hermit crab arthropods). Among the taxa list generated for the pingo site, cnidarians accounted for the largest number of taxa (18), followed by echinoderms (11 taxa) and then chordates (10 taxa). After the frenulates, *Gersemia* corals and spherical *Thenea* sponges were the next most numerous groups of animals. A few different types of commercial species were seen, including Atlantic cod (*Gadus morhua*), Northern shrimp (*Pandalus borealis)*, haddock (*Melanogrammus aeglefinus)* and Greenland halibut (*Hippoglossoides platessoides*). Only one chemosynthesis based species

was seen, the siboglinid frenulate worm, *O. haakonmosbiensis.* This species is also the only animal seen at the site that could possibly be a seep obligate species. However, the generalist lifestyle of frenulates (Hilário et al., 2011; Southward, 2000; Southward et al., 2005) and the debate around the consideration of *O. haakonmosbiensis* as a separate species from the fjord frenulate *Oligobrachia webbi* (Meunier et al., 2010), means that it is possible that, despite the cold seep setting, the entire pingo community consists solely of background benthic species, regardless of whether community members are

chemosynthesis based or conventionally heterotrophic. Nonetheless, the animals at the study site appear to take advantage of, and utilize the seep environment and its inherent characteristics.

In some cases, such as for hard substrate dwelling animals like sponges or corals, the benefits of a seep system on the benthic landscape is evident: it provides hard settlement surfaces, in the form of authigenic carbonates, in a predominantly soft

sediment seafloor (Becker et al., 2009; Cordes et al., 2008; Levin et al., 2015). The advantage of abundant hard settlement substrates likely corresponds with cnidarians and sponges being the most abundant and speciose of the phyla seen at the site. Mobile species, such as *P. borealis* shrimp and ophiuroids were also often seen among carbonates and for these taxa, the advantage of these structures likely lies in the increase in habitat heterogeneity created by them, which can provide shelter or protection (Åström et al. 2017b).

Other taxa might make use of the other major advantage of the seep environment, i.e., autochthonous chemoautotrophic primary production, which tends to exceed the photosynthesis derived, detrital food supply of the surrounding seafloor (Sibuet and Olu-Le Roy, 2002). Indeed, certain taxa appeared to show affinities for chemosynthesis based seep habitats, i.e., the frenulate worms and bacterial mats, both of which constitute the base of the local food chain. One of the snow crabs, for

example, was seen grazing among the bacterial mats. Importantly, this is likely the first record of such behavior in snow crabs, since they are not normally associated with cold seeps. Other crabs in the Majidae family have been seen at cold seeps (Martin and Haney, 2005) and are considered to either be grazers of filamentous bacteria or predators of vesicomyid clams (Barry et al., 1996). While the chemosynthetic members themselves could serve as a food source for certain animals, the combination of high primary production and settlement surfaces together could lead to higher food availability for other, and particularly,



higher order consumers. Siboglinid worms, including frenulates, are known to enhance local infaunal diversity and density (Bernardino et al., 2012), and samples from the pingo study site contained numerous instances of foraminifera, polychaetes and caprellids adhering to the tubes of the worms. Pycnogonids were largely associated with frenulate worms, and mosaics or image transects without frenulates contained the lowest numbers of pycnogonids. *P. borealis* individuals were often present among bacterial mats and frenulates, in addition to often being in and around carbonate concretions. Since these animals are known to be predators and scavengers (Arnaud and Bamber, 1988; Bergström, 2000), the advantage of the frenulate habitat is likely enhanced prey availability. Similarly, certain gastropods were seen perched atop carbonate structures and it is unlikely that the hardness of the concretions, or their sheltering properties are of particular significance for this shelled group of animals. Rather, it is probably the dense colonization of these structures by various animals that accounts for this association with carbonates, since the observed gastropods likely also have predatory or scavenging feeding styles (Aguzzi et al., 2012).

One of the numerically most abundant taxa at the site was the spherical sponge, *Thenea* sp. Individuals did not appear to associate with any seep specific features or habitats. Instead, they were seen on soft sediment and this genus is known to use fascicles of spicules to anchor itself into sediment and mud (Vacelet and Donadey, 1977). Similarly, the onuphid polychaete, *Nothria conchylega* was seen in large numbers at the study site (Table 1), but did not display an affinity for any seep habitat such as carbonates, bacterial mats or frenulate worms. Both *Thenea* sponges and *N. conchylega* are common benthic taxa and their quantities at the study site could simply be due to the site falling within their natural distributional range. On the other hand, the local productivity of the seep system could be beneficial for them and account for their high numbers at the study site (since the increased availability of hard substrate is of no particular consequence to these soft sediment dwelling animals). Indeed spherical sponges (*Stelleta* and *Pseudosuberites* genera) occur in high abundances in New Zealand seeps on the Hikurangi margin, where they are associated with sulfidic sediments and areas of active seepage (Baco et al., 2010; Bowden et al., 2013; Thurber et al., 2010). Similarly, the onuphid polychaete, *Hyalinoecia artifex* has been observed at US Atlantic seeps, where they maintain a carnivorous, epibenthic lifestyle, crawling and dragging their tubes along the seafloor (Meyer et al., 2016). Trails in the sediment were seen behind *N. conchylega* individuals in this study, which is evidence for crawling behavior on the seafloor of this species as well (see Fig. 9). Clearly visible trails associated with *N. conchylega* is of significance since this species has been postulated to exhibit crawling behavior (Budaeva and Paxton, 2013; Hayward and Ryland, 1995, 1995), but to our knowledge, this is the first time such behavior has actually been documented.

Other than food and substrate availability, another possible advantage of the seep environment that could be capitalized upon by the resident animals is a diverse and abundant microbial community, including members that are less abundant in background sediment. For example, seep sediment is dominated by sulfate reducing and sulfur oxidizing bacteria as well as methanotrophs, whereas seafloor sediment from non-seep areas is dominated by more cosmopolitan bacteria (Seabrook et al., 2017). Spherical *Pseudosuberites* sponges from New Zealand seeps are even hypothesized to be chemoautotrophic (Thurber et al., 2010). In general, sponges and corals tend to have a highly diverse bacterial microbiome (Blackall et al., 2015; Bourne



et al., 2016; Vacelet and Donadey, 1977). The dominant members of coral microbiomes are Proteobacteria (particularly Gamma and Alpha) (Bourne et al., 2016; Littman et al., 2009; Rohwer et al., 2002), and Gammaproteobacteria are known to be common members of seep sediment communities (Valentine, 2011), including at HMMV (Lösekann et al., 2007; Niemann et al., 2006). Archaea, including anaerobic methanotrophs and nitrate reducers are also known to associate with corals (Siboni et al., 2008; Wegley et al., 2004), and archaeal anaerobic methanotrophs (known as ANME) are key players in the anaerobic oxidation of methane (AOM) that is so fundamental to seep geochemisty (Boetius and Wenzhöfer, 2013; Knittel et al., 2005; Knittel and Boetius, 2009). Therefore, the pingo seeps could be beneficial for certain species that associate with bacteria because they provide access and exposure to a more diverse array of bacterial strains than is present in the non-seep benthic seafloor.

Importantly, sulfide was not detectable in bottom water or even in the upper 5 cm of the sediment at the pingo site. Therefore, background species are capable of colonizing the site and utilizing its associated resources without the complication of having to avoid sulfide toxicity. It should be noted that though certain associations between animal groups and niches or habitats are visible to a certain extent through this study, confirmation of such associations requires further study.

## 4.2 Factors controlling the distribution of chemosynthesis based community members (frenulates)

We hypothesized that the lack of free gas ebullition at GHP5 was representative of this pingo being substantially different from the other three, in terms of both abiotic and biotic features. As hypothesized, community analysis based on data from the georeferenced images and mosaics indicate that the communities on the three free gas emitting GHPs differ, and cluster separately from those on GHP5 (Fig. 2). The TC25 transect over GHP3 appeared to be an exception, because it clustered with the GHP5 communities (Fig. 2). However, this particular image transect did not contain frenulates, a feature shared with the GHP5 mosaics. Since frenulate abundances were in the order of thousands of individuals, community analyses were also conducted with them excluded which resulted in GHP5 communities forming a distinct group from the other pingo communities. Nonetheless, the most obvious difference between GHP5 communities and the communities on the other pingos was the absence of frenulates.

The lack of frenulates from GHP5 has important ecological implications since they are the only confirmed chemosynthetic animal at the study site. All frenulates have obligate, nutritional symbiotic associations with bacterial endosymbionts (Fisher, 1990; Hilário et al., 2011; Southward, 1982; Southward et al., 2005) and molecular data and electron micrographs suggest that thiotrophy is the dominant metabolic mode for symbionts of *O. haakonmosbiensis* (Lösekann et al., 2008; Pimenov et al., 2000). Thus we expected sediment sulfide concentrations at GHP5 to be lower than those at the other pingos, and too low to sustain the frenulate worms and their symbionts.





Contrary to our expectations, sediment sulfide concentrations at GHP5 were not lower in comparison to GHP3 (Fig. 3A). Sulfide (and sulfate) measurements were only possible in cores from GHP3 and GHP5, but due to the other similarities between GHP3 and the other free gas emitting pingos, we consider sulfide/sulfate profiles of GHP3 to be representative of conditions at GHP1 and GHP2 as well. The sulfide concentrations measured on GHP5 were at the milimolar level, which is likely not limiting with respect to supporting chemosynthesis based fauna of this size. Though the exact sulfide needs of frenulates and *O. haakonmosbiensis* have not been quantified, significantly larger chemoautotrophic symbioses are known to be found in environments with lower in situ concentrations of dissolved sulfide (Decker et al., 2017; Podowski et al., 2010; Sarrazin et al., 1999; Sen et al., 2013; Urcuyo et al., 2003). Therefore, the sediment at GHP5 contains more than enough sulfide to theoretically support *O. haakonmosbiensis*, and yet, they are absent from this pingo.

Therefore, another factor likely accounts for the absence of frenulates on GHP5, overriding the advantage of an abundant energy source to this chemo-obligate worm. Colonization being inhibited by an inadequate larval supply can be eliminated because GHP5 is in the vicinity of the other three pingos. In experiments conducted on *Siboglinum fiordicum* frenulates, only larvae reared in containers with 10 cm of sediment grew well (Bakke, 1974), and in general, soft sediment is considered the preferred substrate of frenulates (Southward, 1999, 2000; Southward et al., 2005). Soft sediment is the primary sediment type at GHP5, therefore a lack of suitable substrate does not explain the absence of frenulates at GHP5 either. The settlement cues for frenulates are not known, but methane and sulfide have been hypothesized to serve as such cues for seep animals in general (Cordes et al., 2010). Only sulfide has been tested experimentally, and was shown to positively correlate with settlement of seep associated polychaetes (Levin et al., 2006). However, sulfide was not detected in the bottom water at any of the study pingos, nor was it present in the first few centimeters of the sediment in any of the cores (Fig. 3). If sulfide serves as a settlement cue for *O. haakonmosbiensis*, then GHP5 is not deficient in this regard either, in comparison to the other pingos.

Nevertheless, GHP5 does differ from the pingos in other respects. The geophysical setting of GHP5 was different, with zones of acoustic blanking below GHPs 1-3 absent beneath GHP5 (Fig. 5). Such regions of acoustic wipe-outs are interpreted as gas saturated sediment. Therefore, a sub-surface gas reservoir is likely connected to GHPs 1-3, which allows for advection of gas through the sediment and up into the water column. The absence of acoustic blanking underneath GHP5 suggests reduced sub-surface gas transport, or alternatively, a deeper barrier for upward gas migration, and subsequently, lower upward methane flux. In accordance with this, sediment methane concentrations were lower at GHP5 in comparison to the other pingos (Fig. 3B), and, gas hydrates were not recovered from GHP5, though they were recovered in cores from the other pingos. Correspondingly, AOM rates would be expected to be lower at GHP5. Comparative AOM rates are not available for the different study pingos, but AOM occurs in concert with sulfate reduction, therefore sulfate fluxes can be used to make inferences about AOM rates. Sulfate reduction can and does take place in the absence of AOM as well (Dale et al., 2008; Hong et al., 2014; Wallmann et al., 2006), therefore it is important to differentiate between AOM and the breakdown of organic


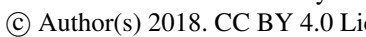


material as being the processes that consume sulfate in a specific system. This can be done by utilizing the stoichiometric relationship of the two sulfate reduction pathways (Claypool et al., 2006; Suess and Whiticar, 1989), specifically, by measuring the ratio of increase in carbon dioxide to the decrease in sulfate. A 2:1 ratio is typical when sulfate is consumed through the anaerobic breakdown of organic matter. This ratio changes to 1:1 when sulfate reduction is linked to AOM. (Masuzawa et al., 1992). Though some of the cores displayed values close to the 2:1 mark in the shallower depths of the sediment, deeper into the sediment, this ratio changes, and approaches the 1:1 correspondence. In most cores, the ratio of inorganic carbon to sulfate consumption was found to be close to 1:1 regardless of depth (both GHP5 and GHP3). The one exception was core 1048 from GHP5, for which, almost all values were closer to the 2:1 ratio. These results indicate that at both pingos 3 and 5, sulfate consumption is linked to AOM, particularly in deeper sediment. With the exception of core 940 (which was on the periphery of GHP3), all the cores from GHP5 exhibited lower sulfate flux rates than those at GHP3 (Table 4). Therefore, our data suggest that AOM rates are lower within GHP5, compared to the other pingos.

Lower AOM rates would result in lower rates of production of two important compounds: hydrogen sulfide, and carbon dioxide (Boetius and Wenzhöfer, 2013; Knittel et al., 2005). Constant replenishment of sulfide, or, a fresh supply, i.e., high sulfide fluxes, might be more important for the frenulate worms than simply high concentrations in the sediment. Additionally, the generation of carbon dioxide might be important for these worms. Because of the carbon fixation activities of their symbionts, frenulates need inorganic carbon, and indeed, RubisCO coding genes have been found in *O. haakonmosbiensis* (Lösekann et al., 2008). Lösekann et al. (2008) hypothesized that *O. haakonmosbiensis* uses carbon dioxide produced either by AOM or the aerobic oxidation of methane. This hypothesis was put forward to explain the extremely negative carbon isotope signatures in *O. haakonmosbiensis* biomarker lipids measured by the authors and by Gebruk et al. (2003), values that had never before been measured in thiotrophic symbioses and which cannot be explained by chemoautotrophic carbon fixation alone, but can be explained by the worms incorporating isotopically light carbon dioxide (DIC) produced by sediment microbes.

The authors further suggested that assimilation of microbial organic carbon (DOC) by the worms could also account for the depleted carbon isotope values. In fact, frenulates differ from other chemosymbiotic siboglinids, such as vestimentiferans, in that they appear to supplement their symbiont provided, chemosynthetic diet with dissolved organic material (Southward et al., 1979, 1981; Southward and Southward, 1970). The larvae of *Siboglinum fiordicum,* was seen, in laboratory experiments, to grow from the larval stage to the juvenile stage over the course of 13 months in which it only used food from yolk reserves and from the surrounding sediment (Bakke, 1977). Furthermore, Dando et al., (2008) noted that in situ, the tubes of frenulates are often turned towards where there are locally high concentrations of reduced organic matter in the sediment.

The dual need for inorganic and organic carbon sources (plus thiotrophic chemoautotrophy) likely results in frenulates in general, and, *O. haakonmosbiensis* specifically, relying heavily on a highly active sediment methanotrophic microbial community. We propose that this is the overriding factor that limits or excludes *O. haakonmosbiensis* from GHP5 despite high





sulfide availability. At the free gas emitting GHPs, the sub-surface gas chimneys lead to constant seepage of methane, which in turn, supports abundant methanotrophic microbes and microbial activity, such as AOM linked to sulfate reduction and sulfide production, thereby providing desirable conditions for *O. haakonmosbiensis.* On the other hand, at GHP5, seepage of methane is low due to the lack of a sub-surface gas chimney. Methane is still present in the sediment, but in lower

concentrations and as a result, methanotrophic microbes are likely less abundant and methanotrophic activity is considerably lower, as evidenced by lower AOM linked sulfate flux rates. A smaller and less active microbial community could mean that GHP5 is deficient in the carbon compounds (either organic or inorganic, or both), required by the worms, and sulfide flux rates might be too low to sustain them as well. A summary of this hypothesis is presented in Fig. 7. Further studies, with a focus on the microbial community and their activity in the sediment will be required to test this hypothesis, though early results indicate

that the microbial community of GHP5 is significantly different from those at the other pingos and that ANMEs make up less of the total microbial community at GHP5 compared to the others (Klasek et al., in prep.). Furthermore, at HMMV, high rates of sulfate reduction and AOM were measured and high numbers of anaerobic methanotrophs were found around the bases of *O. haakonmosbiensis* tubes (Lösekann et al., 2008; Niemann et al., 2006). Similarly,  high sulfate fluxes were measured at the Beaufort Sea *O. haakonmosbiensis* site (sulfate concentration decreased from seawater values to <0.1 mmol within 0.5 m of

the sediment) (Paull et al., 2015). A dependency on sediment methanotrophic microbes and their activity could also account for the absence of *O. haakonmosbiensis* at Arctic hydrothermal vents (Kongsrud and Rapp, 2011; Pedersen et al., 2010), despite its wide distribution across the Arctic because at vents, methane rises quickly  into the water column due to the heat of hydrothermal emissions, thereby reducing the amount available for sediment microbes.

Based on the geochemical data, we propose an additional factor that could contribute to the absence of *O. haakonmosbiensis*

at GHP5. Hong et al. (submitted) found that the porewater from cores taken at GHP5 had a greater contribution from fluids that have undergone high temperature water-rock interactions in comparison to the porewater from the other pingos. The product of such reactions is a higher abundance of dissolved heavy metals, such as lithium, which was measured in higher concentrations in GHP5 porewater compared to the porewater from the other three pingos (Hong et al., submitted). Other metals also tend to accumulate as a result of high temperature rock-water interactions such as barium, cesium and rubidium

(James et al., 2003). Hydrothermal vent members of the siboglinid clade, such as vestimentiferans, have to deal with heavy metal-rich hydrothermal fluids and at least *Riftia pachyptila* is known to contain metal binding proteins that can be used for detoxification purposes (Childress and Fisher, 1992; McMullin et al., 2007). In general, little is known about how vent animals cope with the toxic effect of high levels of heavy metals and frenulates are one of the least studied groups of siboglinids (Hilário et al., 2011). However, frenulates are considered basal within the siboglinid radiation, and a popular hypothesis

regarding siboglinid evolution is that it has been driven by a trend of increased habitat specialization (Halanych et al., 2001; Hilário et al., 2011). Based on this, frenulates would be more generalistic than vestimentiferans in their habitats and could subsequently lack the highly specialized adaptations found in vestimentiferans. Therefore it is possible that *O.*



*haakonmosbiensis* is not equipped to deal with heavy metal-rich fluids. This hypothesis too could explain why the worm is absent both from GHP5 and hydrothermal vent fields in the Arctic.

**4.3 Comparisons to other seep sites**

Few studies have been conducted at other seep sites around the world using imagery and photomosaics to characterize megafaunal communities. Among the few that do, most report substantially lower species/taxa counts compared to the 62 taxa seen (conservatively) in this study: Lessard Pilon et al. (2010) recorded 15 taxa at a Gulf of Mexico seep site (approx.. 2000 m water depth) and 13 taxa were seen in the density study by Olu et al. (2009) at the West African Regab pockmark. In their review, Sibuet and Olu (1998) summarized macrofaunal and megafaunal species counts from all the known seep sites at the time (400 to 6000 m water depth) and found the highest such count to be 42, and on average to be 12. Bowden et al. (2013) on average saw 20 species per site among seeps off New Zealand on the Hikurangi margin (depth range: 744 m -1120 m), although species richness counts were related to the different numbers of images analyzed per site. On the other hand, Rybakova et al. (2013) counted 31 taxa at HMMV (water depth about 1200 m) and Amon et al. (2017) found 36 morphospecies on average across four Caribbean seeps in water depth ranging from 998 to 1600 m.

However, the high diversity at the pingo site is not completely unexpected, given the shallow (less than 400 m) water depth at which it is located. In general, diversity and species richness at cold seeps tend to decrease with increasing water depths (Sibuet and Olu, 1998). This usually applies to symbiont containing species as well, so that shallow seeps have more symbiont containing species than deeper seeps (Sibuet and Olu, 1998). The pingo site could represent a major deviation from this general trend, since it contains only one symbiont containing species. It is possible that one or more of the thyasirid bivalves or the spherical *Thenea* sponges at the site contain symbionts, but even if that were the case, it is unlikely that the numbers of symbiont containing species at the study site would rival the numbers (10-15 species) recorded at shallow sites. It should be noted, though, that only seep sites at water depths of 400 m or more were included in this review, therefore, shallow in this context nonetheless refers to sites often considered the deep sea. In fact, Dando (2010) noted the opposite trend in shallower seeps, i.e., a decrease in numbers of symbiotic species with decreasing water depths. This particular review also does not cover sites at water depths of the study site: it focuses specifically on seeps in water depths of 200 m or less. Sahling et al. (2003) examined depth related trends in seeps along the Sakhalin shelf in the Sea of Okhotsk along a depth gradient of 160 m to 1600 m and observed similar patterns to Dando (2010). These studies and our results illustrate that the relationship between depth and numbers of symbiont containing species at cold seeps is yet to be resolved. It is possible that at least two switching points exist: the shallowest sites have very few symbiont containing species but at some point, possibly 400-500 m, the numbers of symbiont containing species rapidly increases and reaches a maximum, after which, deeper sites again see a drop in the numbers of symbiont containing species. More studies on seeps at intermediate depths, such as those on continental shelves like the present study site will be required to fully explore these trends.





Shallow seeps tend to be populated by a subset of the local, background benthic community (Dando, 2010), and this appears to be true for the pingo study site as well. *O. haakonmosbiensis* is the only possible exception and potential seep obligate species, although Smirnov (2014) reports this species from a muddy bottom site in the Laptev Sea, without any mention of it

being a seep site (and furthermore, *O. haakonmosbiensis* might not be separate from *O. webbi* which has been found in fjords). Background fauna and frenulates, is also what has been observed at other Arctic seep sites, such as pockmarks on the Vestnesa Ridge (Åström et al., 2017b) in the Barents Sea and mud volcanoes in the Beaufort Sea (Paull et al., 2015). At HMMV as well, the community conforms to this basic structure, with the addition of the chemosynthetic monoliferan worm, *Sclerolinum contortum* (Gebruk et al., 2003; Lösekann et al., 2008). These seeps all vary considerably in terms of water depth: 380 m at

the pingo site, 282-740 m for the Beaufort Sea pockmarks, and ~1200 m at both Vestnesa and HMMV. This indicates that in the Arctic, regardless of depth, seeps tend to have similar overall community structure. There is no transition to communities dominated by large, chemosymbiotic seep fauna such as vestimentiferan tubeworms, vesicomyid clams and bathymodioline mussels, seen at about 400 m in seeps at lower latitudes (Sahling et al., 2003). Intriguingly, this separation between Arctic seeps and seeps in other parts of the world with respect to large, chemosymbiotic species, is likely only a modern trend. The

shells of large bodied chemosymbiotic bivalves (thyasirids and vesicomyids) have been recovered in cores from the pingo study site (Åström et al., 2017a), Vestnesa Ridge (Ambrose et al., 2015; Hansen et al., 2017; Sztybor and Rasmussen, 2017) and methane seep deposits on the Gakkel Ridge (Kim et al., 2006) and in the Laptev Sea (Sirenko et al., 1995). Shells from the pingo site have been estimated to be up to 7-14 thousand years old (M. Carroll, unpublished data), and based on the Vestnesa and Gakkel Ridge samples, the extinction event for these animals has been estimated to have taken place around

15,000 years ago (Ambrose et al., 2015; Hansen et al., 2017; Kim et al., 2006; Sztybor and Rasmussen, 2017). This coincides with deglaciation following the Heinrich H1 cold event and the accompanying environmental changes, including extensive releases of methane such as is hypothesized to have created the pingos at the study site (Serov et al., 2017) This could have led to the local extinction of large chemosymbiotic bivalves in the Arctic. Since recolonization has not taken place despite Atlantic water inflow, (and at least Vestnesa and the pingo site fall within the path of the North Atlantic current), the changes

that triggered the presumed demise of the Arctic chemosynthetic bivalves likely persist today. Sztybor and Rasmussen (2017) proposed the drop of bottom water temperatures to sub-zero levels at Vestnesa to be the explanatory factor. However, bottom water temperatures are about 2ºC on average at the pingo site and mean annual bottom water temperature at the Beaufort Sea pockmarks is 0.2ºC (although temperatures as low as -1.1ºC were also recorded, Paull et al., 2015). Vesicomyids of the genus *Isorropodon* have also been sampled at the Nyegga seep site in the Norwegian Sea (Krylova et al., 2011), where bottom water

temperatures are -0.7 ºC (Portnova et al., 2014). These data make it difficult to use modern bottom water thermal regimes as a sufficient reason to explain the death and subsequent lack of recolonization of large chemosynthetic bivalves in the Arctic, although it could play a role. The precise causes for the disappearance and continued absence of large, chemosynthetic bivalves in the Arctic are unclear. Nonetheless, based on the existing data, Arctic seeps appear to form a distinct biogeographical entity,



exhibiting the same, general seep community structure, but one that is different from seep communities in other parts of the world.

Another way in which the pingo site appears to deviate from generalized seep trends relative to lower latitude seeps is with respect to the factors that promote successional progression of the communities. The presence of carbonate slabs on GHP5

indicates that this site could have experienced higher levels of seepage activity in the past and in fact, based on detailed geochemical and geophysical analyses, Hong et al. (submitted, 2017) concluded that this pingo represents a later stage in the geophysical history of these features. Therefore, it is likely that the community on this pingo also represents a later stage in the succession of the pingo seep communities. At present, the chemoautotrophic frenulate worms no longer exist, but the products of seepage, such as carbonate rocks, provide settlement substrates, making this community diverse and densely

colonized. This is similar to what has been observed or predicted at lower latitude seeps, where carbonates and the tubes of vestimentiferans provide a substrate for hard bottomed animals such as sponges or cnidarians (Bergquist et al., 2003; Bowden et al., 2013; Cordes et al., 2005). The major difference is that in these studies and models of succession in lower latitude seeps, a major driving force from an active to a senescent or background fauna dominated community, is a cessation or displacement of fluid flow, often accompanied by a decrease in sediment sulfide concentrations. However, the sediment at GHP5 does not

have lower sediment sulfide concentrations than the sediment hosting the earlier successional communities on the free gas emitting pingos (although methane seepage has likely decreased or even ceased due to the exhaustion of the sub-surface gas chimney). As discussed above, lower latitude seeps at comparable or greater depths are characterized by the presence of more than one type of chemoautotrophic faunal group, each with different geochemical needs and niches. This is not true for the study site, where so far, *O. haakonmosbiensis* alone makes up the entirety of the metazoan chemoautotrophic community. This

difference, of one, compared to multiple chemoautotrophic megafaunal species being present, could account for the pingo site deviating from the trend of successional progression at seeps paralleling changes to sulfide concentration and availability. A highly limited chemosynthetic metazoan community appears to be the norm at Arctic seep sites, therefore, the pattern of successional progression in the absence of depleted sulfide reserves observed at the pingo site, though currently quite unique, could be representative of the Arctic in general, although further studies are needed in order to confirm this.

**4.4 Arctic perspectives**

Among the diverse assemblages of background benthic species present at the pingo site are a number of commercially important species. Atlantic cod (*Gadus morhua*) and the northern shrimp (*Pandalus borealis)* were particularly numerous, but haddock (*Melanogrammus aeglefinus)* and various flat fishes such as Greenland halibut (*Hippoglossoides platessoides*) were also seen. In addition, two individuals of snow crab (*Chionoecetes opilio*) were seen in the images over GHP3 (Table 1).

Commercial species have been observed at seep sites around the world (Baco et al., 2010; Bowden et al., 2013; Grupe et al., 2015; Higgs et al., 2016; Niemann et al., 2013; Sellanes et al., 2008) and the importance of methane seeps to local fisheries is recently gaining attention (Levin et al., 2016). In at least two cases, diets with a significant chemosynthetic component have





been established  for commercial species (Higgs et al., 2016; Niemann et al., 2013). Additionally, seep or site specific characteristics (three dimensional carbonate structures, proximity to oxygen minimum zones, chemical environments excluding predators or parasites) have been hypothesized to account for the enhanced densities of commercial fish species at seeps in comparison to non-seep environments (Levin et al., 2016; Sellanes et al., 2008). At the pingo site, no data currently

exists on whether chemosynthesis derived material constitutes any part of the diets of the observed commercial species, or which features of the seep environment draws them to the location. Nonetheless, species targeted for commercial fishing are abundant at the pingo site. Methane seeps have not been studied intensively in the Arctic and their potential contributions to the Norwegian fishing industry have never been explored. Our results, for the first time, indicate that methane seeps could function as a habitat for multiple economically important species.

Nearby areas not affected by seepage were not imaged during this study, therefore quantitative comparisons between the pingo seeps and the surrounding seafloor with respect to megafauna were not possible. However, the tow cam image transects covered some area outside and to the west of GHP3 (although bacterial mats were seen in this area, so it likely does not constitute a truly non-seep environment). Every individual of every visible taxon was not marked in this area, although the

total number of taxa seen in this area was recorded. In total, 28 taxa were seen in this non-pingo area of 2330 m$^2$, which amounts to 1.2 taxa per 100 m$^2$. This is considerably lower than the richness counts recorded at the pingo mosaics and transects (average 4.1 species per 100 m$^2$, Table 3). Density and abundance data could not be compared with the pingos because this data was not compiled for this area, but the pingos appeared to be more densely colonized than the non-seep area. Qualitative comparisons of faunal abundances and a single comparison of richness counts are not sufficient for drawing robust conclusions

about the differences between pingo-seep communities and the surrounding seabed. Nonetheless, our results suggest the possibility of the pingos creating a biomass and diversity hotspot on the seafloor with respect to megafauna. In fact, this has been suggested (despite a similar absence of quantitative data) for the Concepción seep on the continental slope off Chile (Sellanes et al., 2008), and for HMMV, where Gebruk et al. (2003) noted that the background community appeared to be much 'poorer' than the HMMV community. Åström et al. (2017b) also found higher species richness, biomass and diversity at

Vestnesa Ridge seep sites in comparison to non-seep sites.

Therefore, the seep sites such as the studied pingos could hold ecological significance. The Norwegian government has prioritized protection and mapping of the shelf and areas where coral, sponge, sea pen or other communities of high importance to the Barents Sea-Lofoten ecosystem (Norwegian Ministry of the Environment, 2010). Our results indicate that methane seeps

could constitute one of these communities important to the ecology of the Barents Sea and the Arctic. Sites with gas hydrate reservoirs and seafloor methane emissions appear to be quite extensive along the Arctic shelf in the Barents Sea (Bünz et al., 2012; Sahling et al., 2014; Vadakkepuliyambatta, 2014; Westbrook et al., 2009), therefore, the impact of methane seeps on the larger benthic community could be widespread. However, Arctic shelf seep communities have not been systematically mapped,



nor has their effect on seabed ecosystem dynamics been assessed, therefore we suggest their inclusion into current monitoring, mapping and conservation efforts.

In the Arctic, recognition of and maintenance of diversity hotspots is particularly relevant, because Arctic communities are experiencing substantial disruptions such as species replacements or trophic shifts due to the northward range shifts of many subarctic or temperate species (Degen et al., 2016; Johannesen et al., 2012; Wassmann et al., 2011). It is debatable whether true Arctic biodiversity hotspots exist at all since the meaning of the term sensu strictu refers to areas with high concentrations of endemic species (Myers et al., 2000) and relatively few species are considered as being endemic to the Arctic (Barry et al., 2013). Nonetheless, certain locations in the Arctic do tend to contain elevated numbers of 'true' Arctic species (Barry et al., 2013), and the diversity of the pingo site suggests that shelf cold seeps could fall under this category. Under most circumstances (i.e., in other parts of the world, at lower latitudes), one would not expect a seep site to be affected by the arrival of new species to the surrounding region, because the new arrivals would not be considered capable of successfully establishing themselves within the specialized seep environment. However, the community at the pingos does not contain specialized seep endemics and rather, consists almost entirely of background, benthic species. Furthermore, snow crabs were seen at the pingo site, even grazing on a bacterial mat, and this species has only been seen in the Barents Sea since 1996 (Kuzmin et al., 1998, 1999) and has spread to the north west, reaching west Spitsbergen fjords in 2017 (P. Renaud personal communication). The presence of snow crabs at the site indicates that species new to the area are capable of establishing themselves at the site, which suggests that these communities could experience the same types of upheavals documented at benthic sites along the path of northward migration of southern latitude species (Cochrane et al., 2009; Johannesen et al., 2012; Węsławski et al., 2011).

In short, our results, indicate that the pingo study site and by extension, other shelf seeps could constitute important habitats for multiple commercial species, possibly serve as biomass and diversity hotspots on the seafloor, and could be threatened by climate change induced ecological disturbances. Therefore, it is crucial that benthic mapping efforts and long-term monitoring projects proposed to understand the response of a changing Barents Sea (Barry et al., 2013; Jørgensen et al., 2015) take shelf seep communities into account as well. Since seeps are long lived systems whose effect on the benthos can extend beyond the lifespan of seepage activity itself (Bowden et al., 2013; Cordes et al., 2008; Levin et al., 2016) , careful management policies will need to be drafted, in order to successfully maintain the juxtaposition between maintenance of the seep habitat and its economic exploitation.

### 4.5 Conclusions

Studies focused on the biology and ecology of Arctic cold seeps are rare. Therefore, the natural laboratory conditions of multiple pingos within a limited spatial extent at the study site provided an unprecedented opportunity to study the response of Arctic seep communities, and particularly, chemosynthetic members, to variable physical factors. Our results show that

despite the likelihood of sulfide being the dominant energy source, concentrations of sulfide in the sediment do not necessarily correlate with the presence or absence of the only confirmed chemosynthetic animal at the site, *O. haakonmosbiensis* frenulates. We hypothesize that high sulfide flux, and/or dissolved inorganic or organic carbon produced by microbial methanotrophic activity in the sediment constitute the major carbon source for these worms and small microbial

communities and low methanotrophic activity in the sediment limit the presence of these worms even when sulfide is abundant. Alternatively, site specific conditions, such as deeply sources, heavy metal rich porewater fluids could affect the settlement and development of *O. haakonmosbiensis* at certain locations within the study site. This worm is ubiquitous across Arctic seeps and this is the first time that its distribution could be correlated with variable physical conditions. Overall, the pingo communities are characterized by a diversity of background species and a lack of seep obligate species,

both of which are likely a function of the location of the study site on the shallow (less than 400 m) shelf. This study is the first to document seafloor crawling behavior of *Nothria conchylega* onuphid worms, behavior which has only been hypothesized before. Commercially important fish and crustacean species were seen in large numbers and surprising for a seep site, a species that is relatively new to this area (the snow crab) was seen grazing among bacterial mats. Further investigation of the pingo site and others like it is important to understand shallow water and shelf cold seeps, their effect on

the benthos and their responses or possible susceptibility to a changing and warming Arctic.

**Data availability**

The data are stored at the CAGE Centre for Arctic Gas Hydrate, Environment and Climate's data repository and requests for access should be directed to cage_data@list.uit.no.

**Author contributions**

AS constructed the mosaics, marked animals and conducted the community analyses. EÅ identified most species/taxa. WLH analysed the geochemical data and MW and AP analysed the geophysical data. PS quantified methane concentrations. All authors actively engaged in scientific discussions regarding the study and were all involved in writing the manuscript.

**Competing interests**

The authors declare they have no conflict of interest.





**Acknowledgements**

We would like to thank the captain and crew of *Helmer Hanssen*, the chief scientists and the scientific teams of the cruises from which data for this project was gathered. We are deeply indebted to the tow cam team (Daniel Fornari and Gregory Kurras) and the ROV team (Martin Ludvigsen, Frode Volden, Stein Nornes and Pedro de la Torre), without whom this project would not have been possible. We are also very thankful to Bjørn Runar Olsen and Roy Robertsen, for their patience and expertise with problem solving at sea, Bo Krogh, for helping with the navigation data, and Steinar Iversen for helping with cruise preparations. Hans Tore Rapp, Nataliya Budaeva, Mari Elliertsen, Jon Kongsrud, Irina Zhulay helped with species' identifications, and provided information on their biology, for which we are extremely grateful. Friederike Gründger and Matteus Lindgren analyzed and quantified methane concentrations in cores from 2016 and Arthur Peter Bjørkli prepared Figure 7. We would also like to thank Paul Dando, Karine Olu and Ann Andersen for discussions that helped refine this manuscript, and Diva Amon, Scott Klasek, Ashley Rowden and David Bowden for sharing details of their datasets with us. This study was funded through the Centre for Arctic Gas Hydrate, Environment and Climate (CAGE) and the Research Council of Norway through its Centres of Excellence scheme, project number 223259.

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

Table 1: Total numbers of individuals/aggregations and densities of fauna seen, listed by mosaic/transect. For each taxon, the first number represents the number of individuals, or the number of aggregations in the case of fauna where counting and marking each individual was not possible. Aggregated taxa are marked by a + sign. For these taxa, the number in parentheses refers to the density, calculated based on the area of each mosaic/transect (in the case of frenulates, densities were calculated based on collections and in the case of *Caryophilia* corals, densities were calculated based on selecting a few aggregations per site and counting individuals within aggregations). In the case of bacterial mats, the number in parentheses refers to the density of mats per unit area, since density of individuals of bacteria cannot be counted. Taxa with a * were not used in community analyses. Taxa marked with ^ were seen in images over GHP3 that could not be used due to navigational difficulties associated with that particular lowering of the towed camera.

Table 2: Properties of all the gravity cores taken and analyzed for the study. The measurements taken for each core are listed (sulfide concentration, sulfate concentration, excess inorganic carbon and methane concentration).

Table 3: Diversity indices and taxonomic richness counts (total and normalized for area) for the different mosaics and transects over the study pingos.

Table 4: Sulfate flux rates measured in cores from GHP3 and GHP5.



Figure 1: Location of the gas hydrate pingo (GHP) study site in the Barents Sea and overview of the site (a). Panels b through e are close-up views of the individual pingos. Free gas plumes were observed at all GHPs except GHP5. Image transects are visible as lines over the pingos where each constituent image is shown as a single dark rectangle. Mosaics on GHP5 are shown as larger, irregular sized polygons. Dots represent the locations of the gravity core samples: white represents cores in which

all geochemical measurements were made (sulfide, sulfate, DIC, magnesium, calcium and methane), yellow dots are cores in which all geochemical measurements except methane were made, and purple dots represent cores from which only methane was measured.

Figure 2: Examples of image transects and mosaics used in the study. (a): A portion of the TC25 image transect over GHP5. Individual images positioned in space are shown and close up views of two of the images are shown to the right. (b): ROV

mosaic 3 over GHP5, with a close up view of one of the images used to construct the mosaic.

Figure 3: Dendrograms and MDS plots of the communities seen in the mosaics and transects over the different gas hydrate pingos. The top panel (a) are the results with frenulates included in the community analysis (R = 0.926, p = 0.03) while the bottom panel (b) are the results without frenulates (R = 0.704, p = 0.03) In both cases, a fourth root transformation was applied

to the densities, and Bray-Curtis similarity was used.

Figure 4: Concentrations of sulfide (a) and methane (b) in gravity core samples. Cores from GHP5 are represented with red, dashed lines connecting individual measurements (filled triangles) within the cores. Solid, black lines and hollow circles represent cores from the other GHPs (GHP1, GHP2 and GHP3). Methane data from cores 911, 912, 913, 914, and 920 are

reproduced from Serov et al., 2017 (*The post-glacial response of Arctic Ocean gas hydrates to climatic amelioration*, Proceedings of the National Academy of Sciences). Sulfide data from cores 911, 920, and 940 are reproduced from Hong et al., 2017 (*Seepage from an arctic shallow marine gas hydrate reservoir is insensitive to momentary ocean warming*, Nature Communications).

Figure 5: Excess inorganic carbon ($\Delta CO_2$ + Mg + $\Delta Ca$) plotted against sulfate flux (consumption) in gravity cores from GHP3 and GHP5. The sediment depth range from where measurements were taken in the cores is listed individually for each core in Table 4. The solid lines represent the theoretical relationships for a 2:1 and 1:1 ratio of inorganic carbon:sulfate, where a 2:1 relationship represents use of sulfate by sulfate reducing bacteria in the breakdown of organic matter, and a 1:1 relationship represents sulfate reduction linked to methane oxidation.

Figure 6: Seismic profile of the four study pingos. The transect for the seismic profile is shown in the map of the study site, starting near GHP1 (point a) and ending near GHP3 (point b). Zones of acoustic blanking beneath the three free gas emitting pingos are highlighted with arrows. No such blanking was seen below GHP5.





Figure 7: Sketch of hypothesized processes affecting the distribution of siboglinid frenulates. On the left is a hypothetical pingo where blanking in the seismic data indicates the presence of a sub surface gas chimney which results in high flux of methane into the sediment. This leads to high rates of anaerobic oxidation of methane (AOM) and a highly active and sizeable methanotrophic microbial community. Subsequently, sulfide flux and both inorganic and organic carbon compounds are high in the sediment, which supports frenulate worms. On the right is a pingo where blanking in the seismic data is not observed, leading to lower methane flux in the sediment and therefore, a smaller microbial community and lower AOM rates. Correspondingly, sulfide flux rates and the quantities of inorganic and organic carbon compounds are lower in the sediment, resulting in the absence of frenulates from the system.

Figure 8: Snow crabs (*Chionoecetes opilio*) seen at GHP3. Note the presence of the crab within a microbial mat in the lower image. Green lasers are 20 cm apart.

Figure 9: (a): Examples of the onuphid worm, *Nothria conchylega* (a few individuals are highlighted with black arrows). Note trails in the sediment behind the onuphids, indicating they crawl on the seafloor surface. (b): Examples of *Thenea* sponges (possibly *Thenea valdiviae*). A few individuals are circled in yellow. Green lasers are 20 cm apart.



| Phylum and taxon | Number of individuals/aggregations (density) | | | | | | |
|---|---|---|---|---|---|---|---|
| | TC21 GHP2 | TC18 GHP3 | TC25 GHP1 | TC25 GHP3 | ROV1 GHP5 | ROV2 GHP5 | ROV3 GHP5 |
| **Non-animals** | | | | | | | |
| Bacterial mats*+ | 1078 (0.63) | 1313 (0.96) | 216 (0.27) | 40 (0.09) | 26 (0.03) | 6 (0.01) | 2 (0.01) |
| Carbonates (and other rock features)* | 1558 (0.91) | 93 (0.07) | 4161 (5.28) | 2 (0.004) | 918 (1.16) | 296 (0.46) | 985 (3.08) |
| | | | | | | | |
| **Porifera** | | | | | | | |
| *Phakellia* sp. (Elephant ear sponge) | 5 (0.003) | 0 (0) | 6 (0.01) | 0 (0) | 7 (0.01) | 2 (0.003) | 5 (0.02) |
| *Thenea* sp. (possibly *valdiviae*) | 1381 (0.81) | 772 (0.57) | 74 (0.09) | 265 (0.57) | 102 (0.13) | 103 (0.16) | 47 (0.15) |
| *Stylocordyla borealis* (stalked sponge) | 1 (0.001) | 0 (0) | 2 (0.003) | 1 (0.002) | 1 (0.001) | 0 (0) | 1 (0.003) |
| Unknown species 1 (white) | 0 (0) | 0 (0) | 8 (0.01) | 0 (0) | 5 (0.01) | 0 (0) | 0 (0) |
| Unknown species 2 (encrusting, yellow) | 0 (0) | 0 (0) | 5 (0.01) | 0 (0) | 2 (0.003) | 0 (0) | 41 (0.13) |
| Unknown species 3 (stick sponge) | 0 (0) | 0 (0) | 6 (0.01) | 0 (0) | 0 (0) | 0 (0) | 5 (0.02) |
| Unknown species 4 (encrusting, white) | 78 (0.05) | 15 (0.01) | 187 (0.24) | 1 (0.002) | 200 (0.25) | 264 (0.41) | 374 (1.17) |
| Unknown species 5 (white) | 0 (0) | 0 (0) | 5 (0.01) | 0 (0) | 0 (0) | 0 (0) | 1 (0.003) |
| | | | | | | | |
| **Cnidaria** | | | | | | | |
| *Bolocera* sp. | 185 (0.11) | 96 (0.07) | 197 (0.25) | 4 (0.01) | 137 (0.17) | 77 (0.12) | 149 (0.47) |
| *Caryophillia* sp. (pink)+ | 2 (0.01) | 13 (0.39) | 18 (1.38) | 0 (0) | 5 (0.17) | 6 (0.01) | 4 (2.24) |
| *Caryophillia* sp. (white)+ | 88 (1.94) | 4 (0.001) | 646 (11.56) | 1 (0.02) | 2 (0.08) | 0 (0) | 3 (0.69) |
| *Cerianthus* sp. (soft bottom anemone) | 117 (0.07) | 76 (0.06) | 42 (0.05) | 2 (0.004) | 19 (0.02) | 9 (0.01) | 7 (0.02) |
| *Corymorpha* | 29 (0.02) | 6 (0) | 54 (0.07) | 0 (0) | 16 (0.02) | 10 (0.02) | 31 (0.1) |
| *Difa glomerata* (Cauliflower coral) | 0 (0) | 0 (0) | 6 (0.01) | 0 (0) | 0 (0) | 0 (0) | 1 (0) |
| *Edwardsiidae* | 47 (0.03) | 22 (0.02) | 187 (0.24) | 1 (0) | 7 (0.01) | 0 (0) | 0 (0) |
| *Gersemia* sp. (orange) | 85 (0.05) | 104 (0.08) | 206 (0.26) | 31 (0.07) | 1 (0.001) | 0 (0) | 2 (0.01) |
| *Gersemia sp.* (white) | 1001 (0.58) | 410 (0.3) | 621 (0.79) | 240 (0.51) | 380 (0.48) | 313 (0.49) | 328 (1.03) |
| *Hormathia* sp. | 120 (0.07) | 34 (0.02) | 82 (0.1) | 0 (0) | 65 (0.08) | 58 (0.09) | 0 (0) |
| Juvenile anemones | 189 (0.11) | 351 (0.26) | 404 (0.51) | 199 (0.43) | 118 (0.15) | 5 (0.01) | 120 (0.38) |
| *Liponema multicornis* (Pom pom anemone) | 38 (0.02) | 35 (0.03) | 15 (0.02) | 14 (0.03) | 17 (0.02) | 5 (0.01) | 4 (0.01) |





| | | | | | | | |
|---|---|---|---|---|---|---|---|
| *Lucernaria quadricomis* (stalked jellyfish) | 4 (0.002) | 1 (0.001) | 1 (0.001) | 0 (0) | 0 (0) | 0 (0) | 0 (0) |
| Unknown actinarian 1 (small solitary corals) | 15 (0.01) | 20 (0.01) | 50 (0.06) | 8 (0.02) | 4 (0.01) | 0 (0) | 3 (0.01) |
| Unknown actinarian 2 (bright orange) | 43 (0.03) | 0 (0) | 0 (0) | 0 (0) | 0 (0) | 2 (0.003) | 0 (0) |
| Unknown medusa | 0 (0) | 0 (0) | 13 (0.02) | 0 (0) | 0 (0) | 0 (0) | 0 (0) |
| Unknown octocoral 1 (orange) | 103 (0.06) | 5 (0.004) | 25 (0.03) | 6 (0.01) | 0 (0) | 0 (0) | 1 (0.003) |
| Unknown octocoral 2 (yellow) | 0 (0) | 10 (0.01) | 0 (0) | 0 (0) | 0 (0) | 0 (0) | 0 (0) |
| **Ctenophora** | | | | | | | |
| *Beroe cucumis* | 0 (0) | 0 (0) | 1 (0.001) | 0 (0) | 0 (0) | 0 (0) | 0 (0) |
| **Nemertea** | | | | | | | |
| Nemertean, species unknown | 36 (0.02) | 36 (0.03) | 7 (0.01) | 10 (0.02) | 2 (0.003) | 0 (0) | 0 (0) |
| **Priapulida** | | | | | | | |
| Priapulid, species unknown | 0 (0) | 0 (0) | 1 (0.001) | 0 (0) | 0 (0) | 0 (0) | 0 (0) |
| **Sipuncula** | | | | | | | |
| Sipunculid, species unknown | 81 (0.05) | 53 (0.04) | 53 (0.07) | 0 (0) | 0 (0) | 0 (0) | 0 (0) |
| **Annelida** | | | | | | | |
| *Aphrodita* sp. (sea mouse) | 1 (0.001) | 0 (0) | 0 (0) | 0 (0) | 0 (0) | 0 (0) | 0 (0) |
| *Nothria conchylega* (onuphids)* | 270 (0.16) | 170 (0.12) | 311 (0.39) | 559 (1.2) | 0 (0) | 0 (0) | 0 (0) |
| *Oligobrachia haakonmosbiensis* (siboglinids)+ | 619 (1059.92) | 947 (2144.19) | 339 (671.45) | 0 (0) | 0 (0) | 0 (0) | 0 (0) |
| Unknown species 1*+ | 0 (0) | 15 (N/A) | 11 (N/A) | 0 (0) | 0 (0) | 0 (0) | 0 (0) |
| Unknown species 2* | 4 (0.002) | 0 (0) | 10 (0.01) | 24 (0.05) | 0 (0) | 0 (0) | 0 (0) |
| **Arthropoda** | | | | | | | |
| *Chionoecetes opilio* (snow crab)^ | N/A | N/A | N/A | N/A | N/A | N/A | N/A |
| *Euphausiacea* (krill) | 54 (0.03) | 0 (0) | 0 (0) | 0 (0) | 0 (0) | 0 (0) | 0 (0) |
| *Pandalus borealis* (Northern shrimp) | 359 (0.2) | 155 (0.11) | 227 (0.29) | 38 (0.08) | 277 (0.35) | 59 (0.09) | 34 (0.11) |




| | | | | | | | |
|---|---|---|---|---|---|---|---|
| Pycnogonids | 483 (0.28) | 249 (0.18) | 76 (0.1) | 12 (0.03) | 0 (0) | 0 (0) | 1 (0.003) |
| **Mollusca (/Arthropoda)** | | | | | | | |
| Gastropods/Hermit crabs | 64 (0.04) | 53 (0.04) | 54 (0.07) | 9 (0.02) | 6 (0.01) | 3 (0.005) | 1 (0.003) |
| **Echinodermata** | | | | | | | |
| *Chiridota* sp. | 0 (0) | 0 (0) | 3 (0) | 0 (0) | 0 (0) | 0 (0) | 0 (0) |
| *Cucumaria* sp. | 2 (0.001) | 0 (0) | 0 (0) | 0 (0) | 0 (0) | 0 (0) | 0 (0) |
| *Elpidia* sp. (sea pig) | 0 (0) | 1 (0.001) | 0 (0) | 0 (0) | 1 (0.001) | 1 (0.002) | 0 (0) |
| *Henricia* sp. (pink) | 4 (0.002) | 0 (0) | 0 (0) | 0 (0) | 0 (0) | 0 (0) | 0 (0) |
| *Henricia* sp. (white) | 2 (0.001) | 0 (0) | 0 (0) | 0 (0) | 1 (0.001) | 0 (0) | 1 (0.003) |
| *Henricia* sp. (orange) | 3 (0.002) | 1 (0.001) | 4 (0.01) | 0 (0) | 0 (0) | 1 (0.002) | 0 (0) |
| *Henricia* sp. (yellow) | 0 (0) | 0 (0) | 0 (0) | 0 (0) | 2 (0.003) | 0 (0) | 0 (0) |
| *Holothuridae* (species unknown) | 2 (0.001) | 0 (0) | 0 (0) | 0 (0) | 0 (0) | 0 (0) | 0 (0) |
| *Molpadia borealis* | 1 (0.001) | 2 (0.001) | 0 (0) | 0 (0) | 0 (0) | 0 (0) | 0 (0) |
| Ophiuroids | 123 (0.07) | 106 (0.08) | 208 (0.26) | 0 (0) | 0 (0) | 0 (0) | 9 (0.03) |
| *Poraniomorpha* sp. | 0 (0) | 0 (0) | 1 (0.001) | 0 (0) | 0 (0) | 0 (0) | 0 (0) |
| **Chordata** | | | | | | | |
| *Anarhichas minor* (spotted wolffish)^ | N/A | N/A | N/A | N/A | N/A | N/A | N/A |
| *Gadus morhua* (Atlantic cod) | 335 (0.2) | 16 (0.01) | 0 (0) | 2 (0.004) | 2 (0.003) | 99 (0.16) | 77 (0.24) |
| Gray tunicates+ | 0 (0) | 0 (0) | 8 (0.01) | 0 (0) | 0 (0) | 0 (0) | 0 (0) |
| *Hippoglossoides platessoides* (American plaice) | 5 (0.003) | 5 (0.003) | 1 (0.001) | 0 (0) | 2 (0.003) | 0 (0) | 0 (0) |
| *Leptagonus* sp. (snake blenny) | 0 (0) | 1 (0) | 0 (0) | 0 (0) | 0 (0) | 0 (0) | 0 (0) |
| *Lycodes reticulatus* | 1 (0.001) | 1 (0.001) | 0 (0) | 0 (0) | 1 (0.001) | 0 (0) | 0 (0) |
| *Melanogrammus aeglefinus* (Haddock) | 0 (0) | 0 (0) | 0 (0) | 0 (0) | 0 (0) | 0 (0) | 1 (0.003) |
| *Reinhardtius hippoglossoides* (Greenland halibut) | 3 (0.002) | 0 (0) | 0 (0) | 1 (0.002) | 0 (0) | 0 (0) | 0 (0) |
| Skates | 4 (0.002) | 1 (0.001) | 0 (0) | 1 (0.002) | 1 (0.001) | 1 (0.002) | 1 (0.003) |
| Zoarcids (small) | 1 (0.001) | 4 (0.003) | 1 (0.001) | 0 (0) | 0 (0) | 0 (0) | 0 (0) |





**Others/Unknown**

| | | | | | | | |
|---|---|---|---|---|---|---|---|
| White, possible scaphopod* | 46 (0.03) | 89 (0.07) | 56 (0.07) | 44 (0.09) | 0 (0) | 0 (0) | 0 (0) |



| Core number | GHP number | Year | Measurements taken |
|---|---|---|---|
| 911 | 3 | 2015 | sulfide, sulfate, DIC, methane, calcium, magnesium |
| 912 | 3 | 2015 | methane |
| 913 | 1 | 2015 | methane |
| 914 | 2 | 2015 | methane |
| 920 | 5 | 2015 | sulfide, sulfate, DIC, methane, calcium, magnesium |
| 940 | 3 | 2015 | sulfide, sulfate, DIC, calcium, magnesium |
| 1045 | 3 | 2016 | sulfide, sulfate, DIC, methane, calcium, magnesium |
| 1048 | 5 | 2016 | sulfide, sulfate, DIC, calcium, magnesium |
| 1068 | 5 | 2016 | sulfide, sulfate, DIC, methane, calcium, magnesium |
| 1069 | 5 | 2016 | sulfide, sulfate, DIC, methane, calcium, magnesium |
| 1070 | 5 | 2016 | sulfide, sulfate, DIC, methane, calcium, magnesium |




| Mosaic/Transect | Mosaic/transect area (m²) | Total Richness | Richness/100 m² | Margalef's index (d) | Pielou's evenness (J) | Shannon diversity (H') | Simpson's index (1-λ) |
|---|---|---|---|---|---|---|---|
| with sibolginids included | | | | | | | |
| TC21 GHP 2 | 1714.23 | 41 | 2.4 | 5.74 | 0.007 | 0.026 | 0.006 |
| TC18 GHP 3 | 1363.22 | 33 | 2.4 | 4.17 | 0.003 | 0.009 | 0.002 |
| TC25 GHP 1 | 787.63 | 39 | 5.0 | 5.83 | 0.016 | 0.057 | 0.013 |
| TC25 GHP 3 | 467.56 | 20 | 4.3 | 32.04 | 0.578 | 1.732 | 1.703 |
| ROV1 GHP 5 | 787.99 | 28 | 3.6 | 47.99 | 0.636 | 2.119 | 1.948 |
| ROV2 GHP 5 | 637.52 | 18 | 2.8 | 36.32 | 0.666 | 1.926 | 2.156 |
| ROV3 GHP 5 | 319.50 | 27 | 8.5 | 19.04 | 0.613 | 2.022 | 1.089 |
| | | | | | | | |
| without siboglinids | | | | | | | |
| TC21 GHP 2 | 1714.23 | 40 | 2.3 | 35.79 | 0.673 | 2.483 | 1.301 |
| TC18 GHP 3 | 1363.22 | 32 | 2.3 | 46.43 | 0.684 | 2.37 | 1.756 |
| TC25 GHP 1 | 787.63 | 38 | 4.8 | 24.78 | 0.726 | 2.641 | 1.159 |
| TC25 GHP 3 | 467.56 | 20 | 4.3 | 32.04 | 0.578 | 1.732 | 1.703 |
| ROV1 GHP 5 | 787.99 | 28 | 3.6 | 47.99 | 0.636 | 2.119 | 1.948 |
| ROV2 GHP 5 | 637.52 | 18 | 2.8 | 36.32 | 0.666 | 1.926 | 2.156 |
| ROV3 GHP 5 | 319.50 | 27 | 8.5 | 19.04 | 0.613 | 2.022 | 1.089 |





| Core number | GHP number | Sulfate flux (mole/m$^2$/day) | Depth range for measurements (cm) |
|---|---|---|---|
| 911 | 3 | 2.08 | 15-74 |
| 1045 | 3 | 1.27 | 10-110 |
| 940 | 3 | 0.31 | 5-313 |
| 920 | 5 | 0.37 | 10-240 |
| 1048 | 5 | 0.12 | 10-324 |
| 1068 | 5 | 0.90 | 12-308 |
| 1069 | 5 | 0.58 | 8-206 |
| 1070 | 5 | 0.90 | 8-266 |











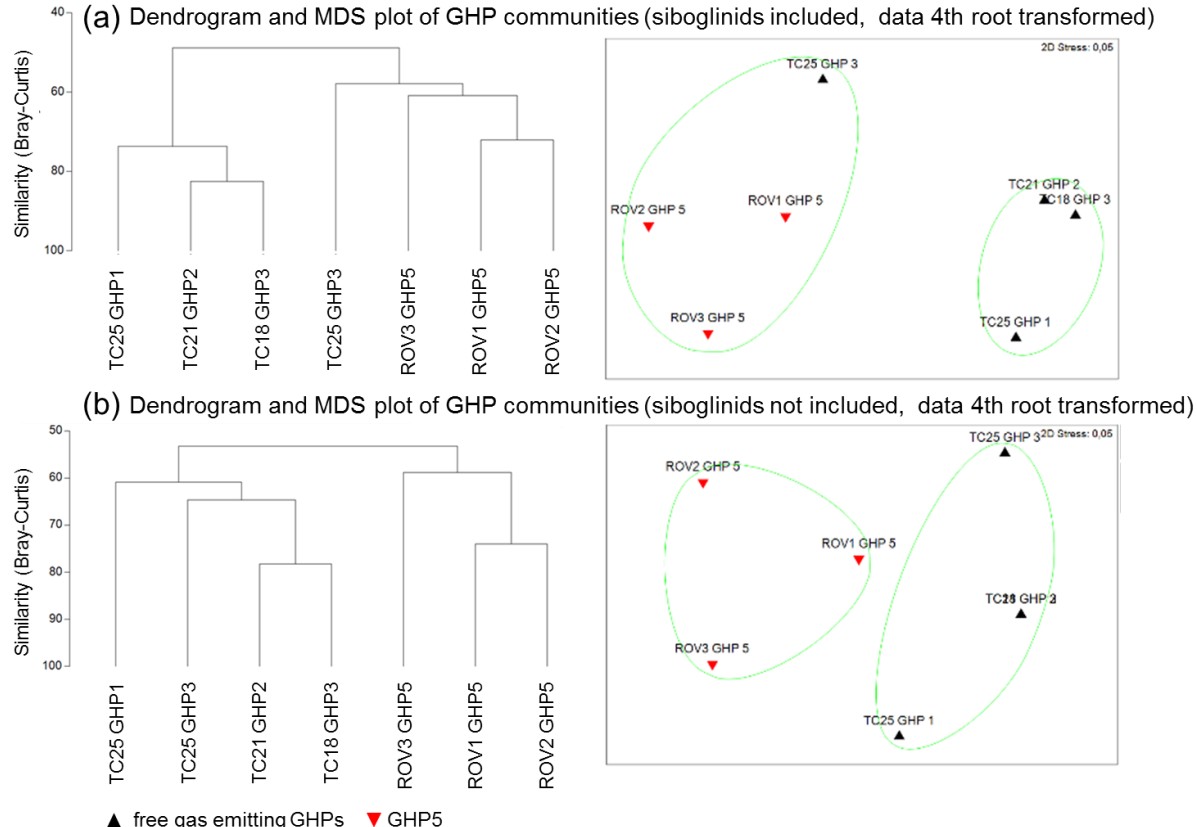





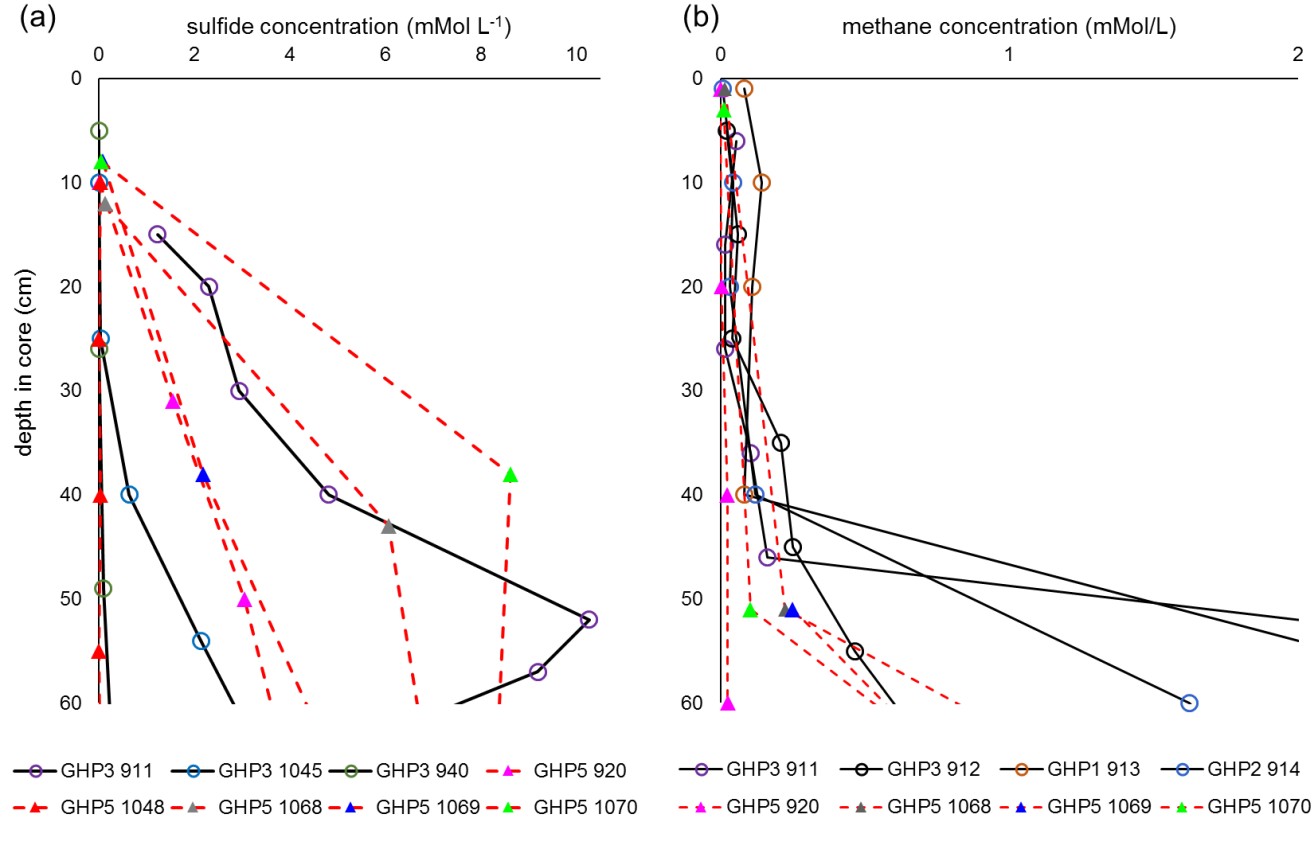



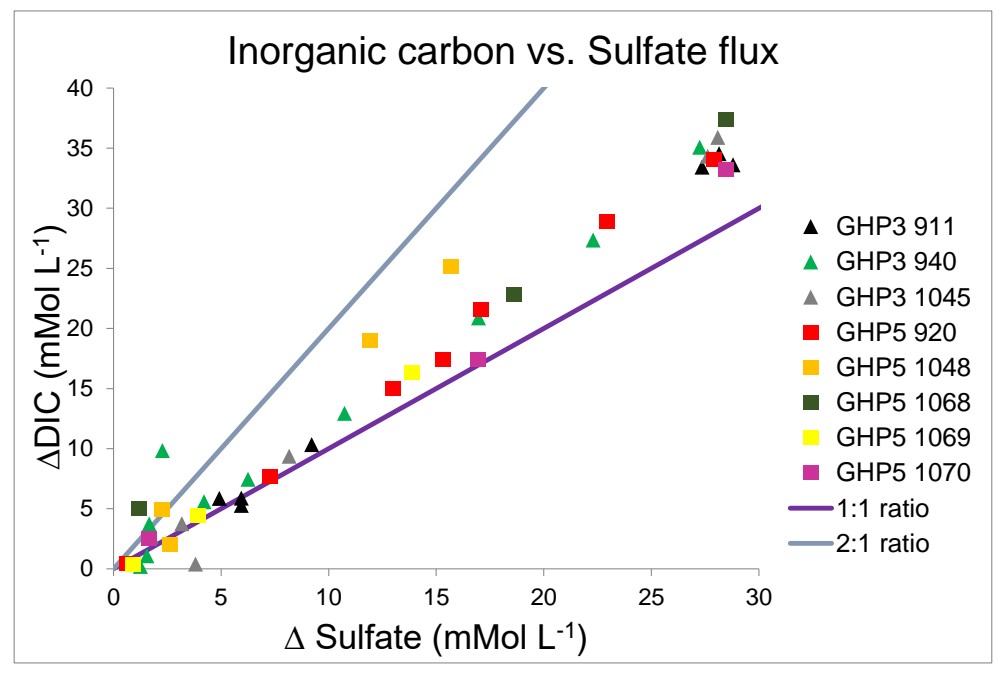





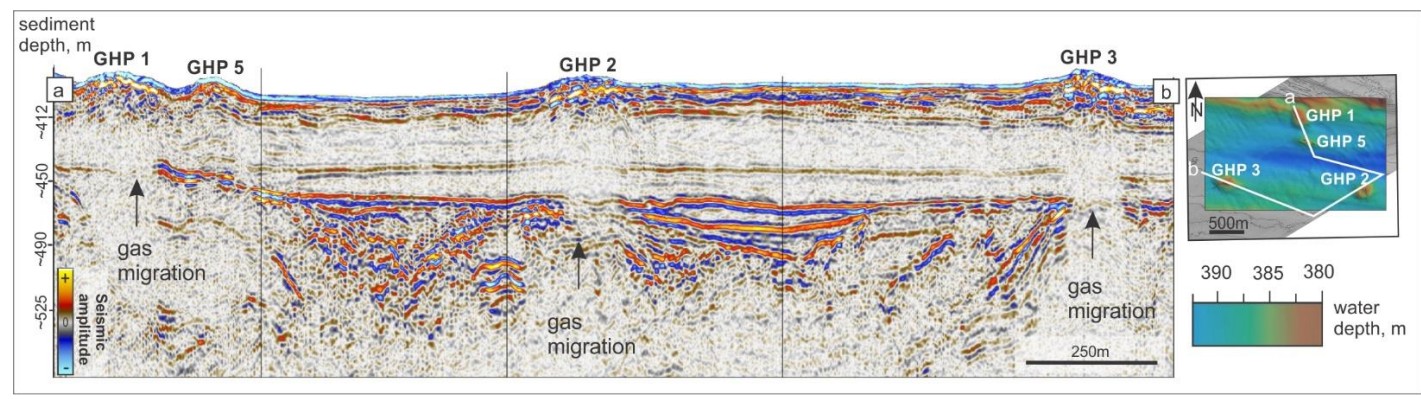





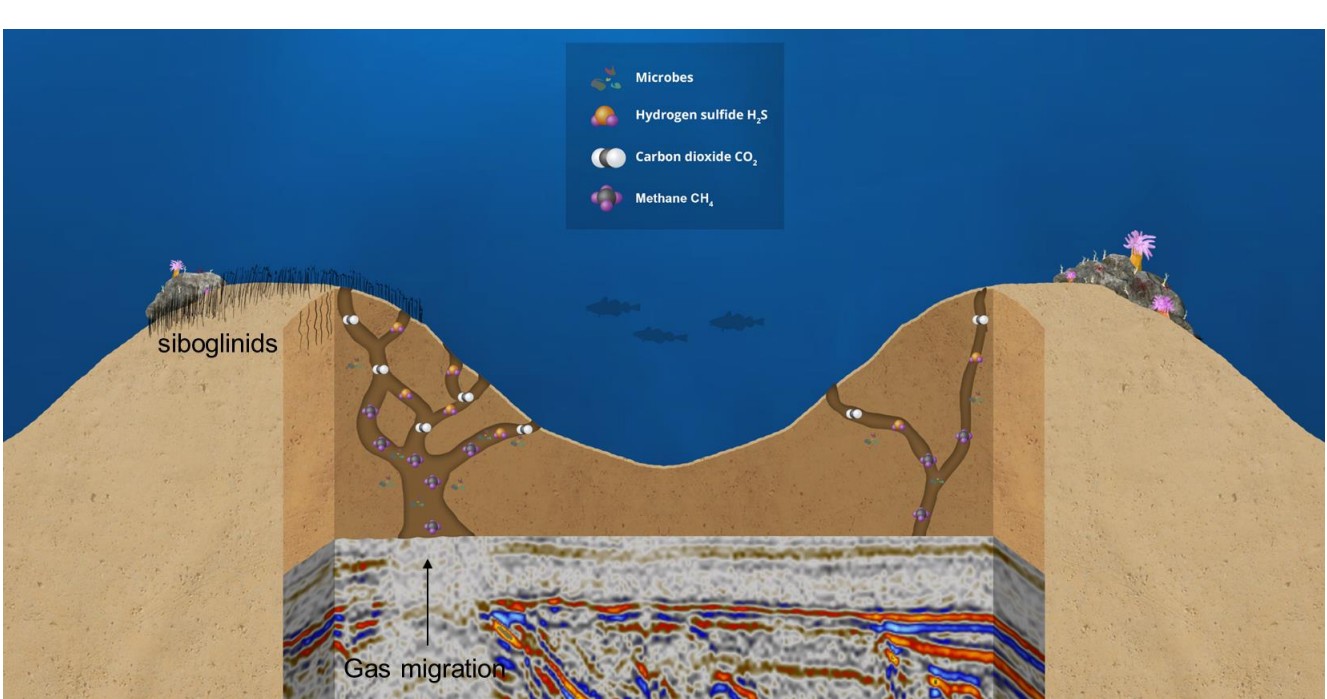





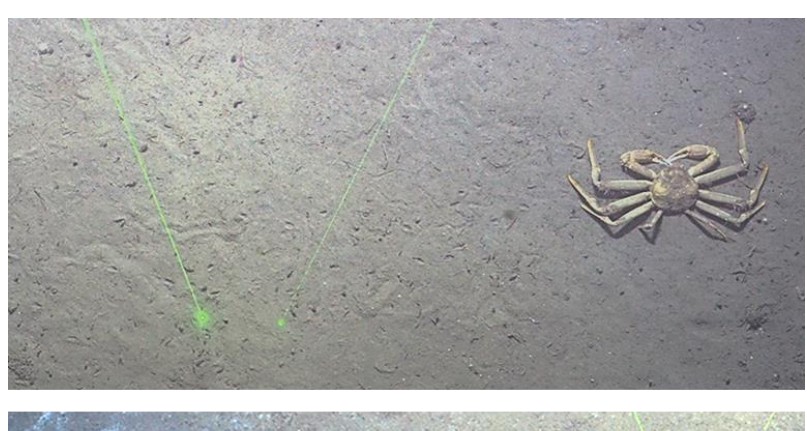

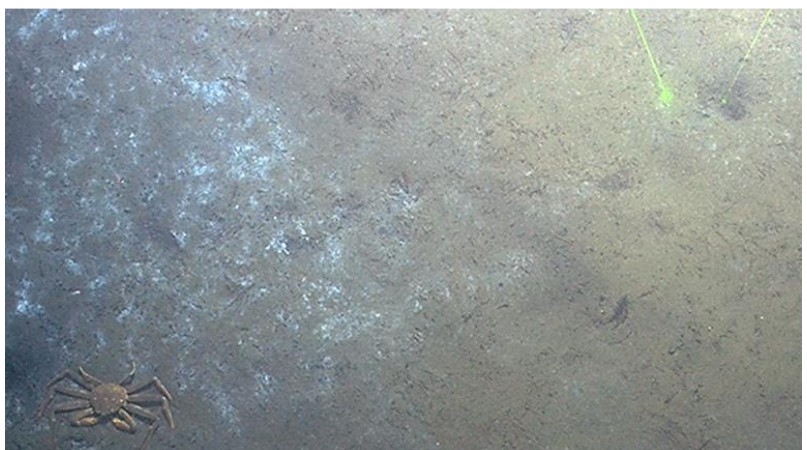





