# Peer review of "Geophysical and geochemical controls on the megafaunal community of a high Arctic cold seep"

_Biogeosciences, 2017_

## Referee Comment (RC1) · P. R. Dando (Referee) · 9 Mar 2018

P. R. Dando

pdando@mba.ac.uk

**General Comments**

This is a well-written manuscript describing the geochemistry, physical habitat and fauna at a series of methane-seeps in the Arctic. The authors relate changes in faunal distribution, from photographic images, to the local chemical and physical environment and discuss the micro-habitats available at such seeps. The manuscript would be improved if it were possible to make comparisons with the background fauna away from seep influence. There are a number of problems with the analytical procedures, presentation and interpretation of results as described below.

**Specific Comments**

Methane

Since the study is based on methane seepage it is essential to have some reliable measurements of methane concentrations available to the biota. Unfortunately the method described for methane analysis does not measure "dissolved pore water methane", as stated, but a mixture of free methane plus methane sorbed to the sediment and released by the sodium hydroxide addition (Ertefai et al. 2010). Since authigenic carbonate is present, the concentration of the sorbed methane can be up to two orders of magnitude higher than the dissolved methane (Ijir et al. 2009). It is unclear why pore water obtained from the rhizons was not used for on-board methane analysis. There is no information available, to my knowledge, on the extent to which sorbed methane is available to the biota. Thus comparisons between sites based on methane availability are thus not valid.

Sulphide

Dissolved reduced sulphur species are utilised by chemoautotrophic free-living and symbiotic bacteria as an energy source. The concentration of dissolved "sulphide" ($H_2S + HS^- + S^=$) and thiosulphate is thus an important measurement. The authors state that sulphide was below the detection limit in the bottom water and in the upper "few" cm of the sediment of all the cores. However, sulphide must have been present to support the bacterial mats visible on the surface. The detection limit is not given but since the lowest standard used in the assay was 40 µM (Hong et al., 2017) the method described may not have been able to detect concentrations of a few µM. Many thiotrophic symbiotic associations exist in sediments with dissolved sulphide concentrations of < 1 µM. It is very difficult to prevent oxidation of low concentrations of sulphide in pore water and since the samples were not analysed immediately it is probable that oxidation occurred during preservation and storage.

H$_2$S would have been carried into the upper sediment and the water column in the methane bubbles as well as in the associated water plume (Reeburgh 2009, Dando et al. 1994a). In addition, the drawdown of seawater induced by the rising methane bubbles (O'Hara et al. 1995, Zimmermann et al. 1997) would have locally generated reduced sulphur species from iron sulphides within the sediment (Dando et al. 1994b) as well as producing a halo of less reducing areas surrounding the bubble outlets.

Identification of biota

The statements "Visible fauna (at least a few centimeters across) were identified" (p5) and the statement in the following paragraph that "Numerous individuals of siboglinid worms were seen", appear contradictory. *Oligobrachia haakonmosbiensis* is large for a frenulate, with a tube diameter of 0.6-0.9 mm (Smirnov, 2014). It would be useful to have a high resolution image, perhaps as a

supplementary file,  to show how these individual siboglinids were visible in the photographs, since a lot of the Discussion is based on their presence or absence.

Most of the identifications relied on interpretations of images taken by a towed camera with a resolution of 16 million pixels and with stereo cameras mounted on a ROV with a resolution of 1.4 million pixels.  We are not told the respective field of views photographed by these cameras so it is not possible to estimate the respective resolutions.  It would help interpretation if the authors would calculate the sizes that the respective pixels represent.  Rough calculations, from the dimensions given in Figure 2, suggest that the pixel size in images from the stereo cameras may have been inadequate to resolve smaller organisms, such as Oligobrachia tubes, 0.6-0.9 mm in diameter (Smirnov, 2014) unless they occurred in clumps. The core samples at GHP 5 were taken around the periphery of the pingo (Figure 1) so that it is not possible to deduce from these that *Oligobrachia* was absent from pingo 5.

Another problem in comparing tow cam and ROV pictures is that the ROV imaging was always from fairly discrete areas on pingo 5 while the tow cams were transects covering from the outside into the centre of the pingo (Figure 1).  This might explain why Nothria, for example, was identified in all the tow cams but not in the ROV pictures.  The tow cam epifaunal data, presented in Table 1, should therefore be divided into "on pingo" and "off pingo" sections.  The reason that TC25 GHP3 clusters with the GHP5 ROV camera tracks is probably because tow cam 25 has the greatest proportion of off-pingo track of any of the tow-cams.

The frenulates observed were identified from specimens in the core samples and density estimates for them were calculated from the densities observed in the cores.  It would have helped the interpretations if information had been provided on the depth they reached in the sediment.  This may be site specific, since the penetration depth of a species has been shown to vary between cores (Dando et al. 2008).  *Oligobrachia haakonmosbiensis* were reported penetrating the sediment to a depth of 55 cm at the Hîkon Mosby mud volcano (Lösekann et al. 2007).  It would also be helpful to know whether other macrofauna were recovered from the cores, since most faunal species with chemoautotrophic bacteria at shallower seeps are infaunal and would not show on surface photographs.  An example is at a methane seep at 170 m depth in the N. Sea where 3 such species were found living within the sediment and shells of a fourth, the bivalve *Lucinoma borealis*, were recovered (Dando et al. 1991, Dando 2001):  no epifauna with chemoautotrophic symbionts were observed.  Many frenulates have tubes completely buried within the sediment; thus chemosynthesis is probably more common at the pingo sites than this study of mainly epifauna suggests.

Discussion

On discussing the distribution of the frenulate *Oligobrachia*, the authors wrote:  "the image transects containing siboglinid frenulates were much less even in terms of species abundances compared to the transect and mosaics which did not contain any frenulates".  This would be expected since it has been shown that, on the Rockall slope, frenulate distribution did not cluster with most other taxa and there was an inverse relationship between frenulate density and the density of other benthos (Dando et al. 2008).  This was explained because sediment disturbance by other organisms would increase sulphide oxidation and displace, or bury, the thin tubes of the frenulates.  It should be noted that in the latter study most of the frenulates had a low abundance and none of the tubes projected from the sediment, if they did at all, as far as those of *Oligobrachia haakonmosbiensis* and thus would not provide an epifaunal habitat.  In the one obligate, methane seep frenulate species that occurred in high densities, *Siboglinum poseidoni* (Dando et al. 1994c), no epifauna were noted between or above the projecting tubes.

The authors consider that chemoautotrophic primary production at the pingos might exceed the photosynthetic primary production reaching the sea floor.  The examples they cite are from deeper water, where less photosynthetic production reaches the sea floor.  This is unlikely to be true at 400 m where much more photosynthetic production will reach the seabed.  As shown in a comparative study (Bernadino et al. 2012), the isotopic difference between background and seep fauna was much lower at the Eel River seeps (250-500m) than at deeper seeps at 770 m depth and deeper.  Isotopic evidence of

food inputs is needed to support the authors' hypothesis. Since methane solubility increases with pressure there will, potentially, also be more methane available to the biota at deeper sites.

Regarding sulphide in the upper sediment, p.14 line 11: only 1 measurement at 5 cm depth is shown in Fig. 4 (for core 1045, off the edge of pingo 3). The exact value for this sample is not shown. However, a single measurement does not justify the statement that "sulfide was not detectable - - - even in the upper 5 cm of the sediment at the pingo site". Should this read "sites"?

As mentioned above, it is probable that sulphide was present at significant concentrations for the biota, including the bacterial mats at the surface, but was not detected using the stated analytical procedure. Serov et al. (2017 Fig. S2C) shows a picture from one of the pingos with a white bacteria mat, presumably of sulphur-oxidising bacteria, on top of "tubeworms" that project approximately 4 cm above the sediment, if the scale on the photograph is correct. If these are sulphur-oxidising bacteria then sulphide or thiosulphate must be present in the water column. The "tubeworms" are approximately 10 mm across, measured against the scale on the photograph, and thus cannot be Oligobrachia.

P14, line 21 and subsequent text: "this particular image transect did not contain frenulates." *Oligobrachia haakonmosbiensis* havs large tubes for a frenulate, many species have tubes 100 μM or less across and would not be visible if they did project above the surface, although many species are completely buried within the sediment. More than one species frequently occur in the same core sample (Dando et al. 2008) so that it is not possible to state that frenulates were absent. To be correct you should write that "this transect did not show any 'visible' frenulates".

GHP5 gas release: in many submarine seeps gas is only released at low tide when slight differences in bottom pressure cause the sub-surface gas volume to increase. At other frequently visited methane seep sites, such as the Scanner pockmark, continuous gas release may, or may not, be present during any given cruise. In the absence of data on the tidal conditions when observations were made over GHP5 it is not possible to state that gas was not released from this pingo. The enhanced reflectors below GHP5 indicate subsurface gas and, on enlargement of Figure 6, it is possible to see a small gas "chimney" towards the edge of the pingo (see Fig 6 section), although this is considerably smaller than the chimneys below the other pingos. Core 920 on the edge of GHP 5 contains methane of thermogenic origin (Serov 2017, Table S1), implying a deep source for the gas. Active release of methane from the sediment will channel the methane into the higher porosity release channels. The sediment at the sides of these channels will have a low methane concentration, due to the down-flow of seawater from the sediment surface (Dando et al. 1994a, O'Hara et al. 1995). Thus it is not correct to argue that methane release will stimulate overall sulphate reduction and methanotrophy in a seeping pingo when compared to a non-seeping pingo with a high sediment methane concentration (p17 first paragraph). Microbes may also be removed from the system by the rising fluids.

Sulphate reduction (p16): "In most cores, the ratio of inorganic carbon to sulfate consumption was found to be close to 1:1 regardless of depth (both GHP5 and GHP3). The one exception was core 1048 from GHP5, for which, almost all values were closer to the 2:1 ratio." Core 1048 is shown in Figure 1 to be the furthest away from any pingo, i.e. it is in background sediment. Thus it should be no surprise that in this core sulphate reduction is not dependent upon the presence of methane.

"The dual need for inorganic and organic carbon sources (plus thiotrophic chemoautotrophy) likely results in frenulates **in general**, and, *O. haakonmosbiensis* specifically, relying heavily on a highly active sediment methanotrophic microbial community"(p16 line 31). This is not true as a general statement. Dando et al. (2011), in a study of the relationship between 10 species of frenulates and the chemistry of their habitat, found that, with the exception of 1 obligate methane seep species, all occupied sediments where the dissolved methane concentration was < 1 μM.

"On the other hand, at GHP5, seepage of methane is low due to the lack of a sub-surface gas chimney. Methane is still present in the sediment, but in lower concentrations and as a result, methanotrophic microbes are likely less abundant and methanotrophic activity is considerably lower, as evidenced by lower AOM linked sulfate flux rates" p17. As mentioned earlier the authors do not know the concentration of available methane in the sediment and hence cannot make such comparisons

regarding different methane concentrations at different sites. The values in Figure 4 may just equate to the amount of authigenic carbonate in the samples. A small gas chimney appears to be visible below GHP5 in Figure 6.

The discussion regarding hydrothermal vents is not very appropriate for this paper with respect to O. *haakonmosbiensis*. This is a cold-water species that penetrates approximately 0.5 m into the sediment. At vent sites the temperature within the sediment would, almost certainly, be lethal for the species.

Although *O. haakonmosbiensis* was the only metazoan with chemoautotrophic symbionts found, it does not mean that it was the only one present, since the infauna, where, for example, other frenulates and thyasirid and lucinid bivalves might be expected, was not studied. It is therefore also not correct to state that "the community at the pingos does not contain specialized seep endemics" (p22 line 13) since the infauna were not studied and *O. haakonmosbiensis*, if distinct from *O. webbi*, is probably a seep obligate species. "Endemic" is incorrect in this context since it refers to geographic regions, not habitats.

Figure 1

It would aid interpretation if the positions of the observed gas flares were pinpointed in figures b-e.

Figure 2, 8 & 9

These would benefit from scales in the camera pictures, since the laser spots, when present, are difficult to see.

Figure 3

Figure 3b has TC21 and TC18GHP3 plotted on top of each other, including the labels, so it is not clear what this point represents.

Figure 4

The lines after the final points appear to be extrapolated randomly. If this is because the graphs are part of larger ones and have been cut off at 60 cm then it would be sensible to give the depth and values of the next points in parenthesis at the end of the lines. The coloured open circles are not well differentiated at the scale of the Figure and should be replaced by coloured filled circles to differentiate the cores.

Figure 6

I think the vertical scale is m depth below the sea surface and not sediment depth. Fig, 6b is too small to be useful without enlargement.

Discussion

The term "megafaunal taxa" is used in the Discussion. Megafauna are large animals such as cetaceans and large fish. The correct term for the taxa observed is "macrofauna"

P14 lines 20 & 21, Fig. 2 should read Fig. 3

**References**

Dando, P. R., Austen, M. C., Burke, R. J., Kendall, M. A., Kennicutt, M. C., Judd, A. G., et al. (1991). Ecology of a North Sea pockmark with an active methane seep. Marine Ecology Progress Series, 70: 49-63.

Dando, P. R., Jensen, P., O'Hara, S. C. M., Niven, S. J., Schmaljohann, R., Schuster, U., et al. (1994a). The effects of methane seepage at an intertidal / shallow subtidal site on the shore of the Kattegat, Vendsyssel, Denmark. Bulletin of the Geological Society of Denmark, 41: 65-79

Dando, P. R., Ridgway, S. A., & Spiro, B. (1994b). Sulphide 'mining' by lucinid bivalve molluscs: demonstrated by stable sulphur isotope measurements and experimental models. Marine Ecology Proress Series, 107: 169-175.

Dando, P. R., Bussmann, I., Niven, S. J., O'Hara, S. C. M., Schmaljohann, R., & Taylor, L. J. (1994). A methane seep area in the Skagerrak, the habitat of the Pogonophore, Siboglinum poseidoni, and the bivalve mollusc Thyasira sarsi. Marine Ecology Progress Series, 107: 157-167

Dando, P. R. (2001). A review of pockmarks in the UK part of the North Sea, with particular respect to their biology. Technical Report TR 001 Strategic Environmental Assessment ⊥ SEA2. London: Department of Trade and Industry.

Dando, P. R., Southward, A. J., Southward, E. C., Lamont, P., & Harvey, R. (2008). Interactions between sediment chemistry and frenulate pogonophores (Annelida) in the north-east Atlantic. Deep-Sea Research I, 55: 966-996

Ertefai, T. F, Heuer, V B, Prieto-Mollar, X., Vogt, C., Sylva, S. P, Seewald, J. S., Hinrichs, K-U. (2010) The biogeochemistry of sorbed methane in marine sediments. Geochimica et Cosmochimica Acta, 74: 6033-6048

Hong, W.-L., Torres, M. E., Carroll, J., CrÄmiÅre, A., Panieri, G., Yao, H. and Serov, P. (2017) Seepage from an arctic shallow marine gas hydrate reservoir is insensitive to momentary ocean warming, Nature Communications, 8, ncomms15745, doi:10.1038/ncomms15745

Ijir, A. , Tsunogai, U. Gamo, T., Nakagawa, F., Sakamoto, T.,
Saito, S. (2009) Enrichment of adsorbed methane in authigenic carbonate concretions of the Japan Trench. Geo-Marine Letters, 29:301⊥308

Lösekann, T., Knittel, K., Nadalig, T., Fuchs, B., Niemann, H., Boetius, A. and Amann, R. (2007) Diversity and Abundance of aerobic and anaerobic methane oxidizers at the Haakon Mosby Mud Volcano, Barents Sea, Applied and Environmental Microbiology, 73: 3348⊥ 3362

Reeburgh W. S. (1969) Observations of gases in Chesapeake Bay sediments. Limnology and Oceanography, 14: 368-375.

Serov, P., Vadakkepuliyambatta, S., Mienert, J., Patton, H., Portnov, A., Silyakova, A., Panieri, G., Carroll, M. L., Carroll, J., Andreassen, K. and Hubbard, A. (2017). Postglacial response of Arctic Ocean gas hydrates to climatic amelioration. Proceedings of the National Academy of Sciences, 114: 6215⊥ 6220

Smirnov, R. V. (2014). A revision of the Oligobrachiidae (Annelida: Pogonophora), with notes on the morphology and distribution of Oligobrachia haakonmosbiensis Smirnov. Marine Biology Research, 10: 972-982

Zimmermann, S., Hughes, R. G., & Flugel, H. J. (1997). The effect of methane seepage on the spatial distribution of oxygen and dissolved sulphide within a muddy sediment. Marine Geology, 137: 149-157

Figure 6 section

[Figure]

---

## Referee Comment (RC2) · P. R. Dando (Referee) · 20 Mar 2018

Relating the geochemistry to the species distribution of the macrofauna at seeps is always very difficult. Concentrations of possible energy sources, both in the surface layers, and within the sediment depth to which most macrofauna occur, can change by an order of magnitude or more over a distance of 10-50 cm from the seep outlet. In this study the positions of the cores with respect to the camera tracks are not that well constrained.

Core 920 is shown here as NE of GHP5 while in Hong et al. (2017) it is shown as SE of GHP5. My earlier comment regarding false dissolved methane readings by adding

equal volumes of 1 M sodium hydroxide to samples for the analysis of free methane was influenced by my own early studies in which samples were not analysed at sea and had a longer storage time in contact with NaOH. I accept that a short exposure may not bias the results to the extent I believed, although it would be good to have results from an untreated pore water sample for comparison. Indeed Ertefai et al. also used 1 M NaOH treatment before measuring free methane and then continued the treatment for a longer period to measure adsorbed methane. Timing is everything!

There is at least one paper on free methane concentrations in sediments off Spitzbergen, that did not use sodium hydroxide pre-treatment for dissolved methane measurements, and should be cited Kneis et al. (2004). These authors analysed 26 sediment samples between 15 and 30 cm depth and found a thermogenic methane signature, d13C of -50.8 (mean) in the adsorbed methane but a d13C of -65.2 (mean) in the free methane. The free methane concentrations, 0.5 – 5.5 micromol/litre were lower than measurements from a similar sediment depths in the pingo areas, 6-330 micromol/litre (Serov et al. 2017). It was suggested, Kneis et al. (2004), that the adsorbed methane was not available to the microbes and that the free methane was probably a mixture from both thermogenic and biogenic sources. However, the methane isotope data reported from the pingo area (Serov et al. 2017) was taken from a greater sediment depth so that a direct comparison of the sources of the free methane cannot be made.

Ertefai, T. F, Heuer, V B, Prieto-Mollar, X., Vogt, C., Sylva, S. P, Seewald, J. S., Hinrichs, K-U. (2010). The biogeochemistry of sorbed methane in marine sediments. Geochimica et Cosmochimica Acta, 74: 6033-6048

Knies, J., Damm, E., Gutt, J., Mann, U., Pinturier, L. (2004) Near-surface hydrocarbon anomalies in shelf sediments off Spitsbergen: Evidences for past seepages. Geophysics, Geochemistry, Geosytems 15: Q06003, doi:10.1029/2003GC000687

Serov, P., Vadakkepuliyambatta, S., Mienert, J., Patton, H., Portnov, A., Silyakova, A., Panieri, G., Carroll, M. L., Carroll, J., Andreassen, K. and Hubbard, A. (2017). Postglacial response of Arctic Ocean gas hydrates to climatic amelioration. Proceedings of the National Academy of Sciences, 114: 6215-6220.

---

## Author Comment (AC1) · 24 Mar 2018

P. R. Dando
pdando@mba.ac.uk

**General Comments**
This is a well-written manuscript describing the geochemistry, physical habitat and fauna at a series of methane-seeps in the Arctic. The authors relate changes in faunal distribution, from photographic images, to the local chemical and physical environment and discuss the micro-habitats available at such seeps. The manuscript would be improved if it were possible to make comparisons with the background fauna away from seep influence. There are a number of problems with the analytical procedures, presentation and interpretation of results as described below.

> Response: Thank you for appreciating the premise of the manuscript. Unfortunately, comparisons with background fauna away from seep influence is only possible qualitatively, and this is the approach we took in the manuscript. More details on this are below.

**Specific Comments**

Methane
Since the study is based on methane seepage it is essential to have some reliable measurements of methane concentrations available to the biota. Unfortunately the method described for methane analysis does not measure "dissolved pore water methane", as stated, but a mixture of free methane plus methane sorbed to the sediment and released by the sodium hydroxide addition (Ertefai et al. 2010). Since authigenic carbonate is present, the concentration of the sorbed methane can be up to two orders of magnitude higher than the dissolved methane (Ijir et al. 2009). It is unclear why pore water obtained from the rhizons was not used for on-board methane analysis. There is no information available, to my knowledge, on the extent to which sorbed methane is available to the biota. Thus comparisons between sites based on methane availability are thus not valid.

> Response: We have to disagree with the reviewer saying that what we measured is "…a mixture of free methane plus methane sorbed to the sediment…" therefore claiming absence of any dissolved gas in our samples. Methane is soluble in water and is always present in pore water samples of marine sediments typically demonstrating 50-80 % porosity (Abrams et al., 2017).
> It is possible that some amount of free and sorbed gas is also present in the bottom sediments in situ and some occasional desorption occurred due to NaOH in our samples. Portions of methane purposefully extracted with alkaline technique from sediments at active thermogenic seepage locations are essentially unknown because (1) petroleum exploration prioritizes acidic extraction and vacuum desorption methods, and (2) existing works on alkaline extraction focus on "..environments that are not influenced by thermogenic processes." (Ertefai et al., 2010) and not related to visible seabed seepage. Moreover, Ertefai et al., 2010 report inaccuracy of their sorbed gas analyses from non-seep sites of as much as 65%, implying difficulties during even targeted extraction efforts. During the analytical stage of our work we did not aim to extract any sorbed gases, therefore, we did not apply any long-duration techniques for mechanical disintegration of clayey assemblages (orbital shakers, mills, etc.).

The publication by Ijir et al., suggested by Dr. Dando states that content of adsorbed gas is twice higher in carbonate concretions compared to surrounding sediments. In our work we collected and analyzed the samples that are by all means equal to what Ijir et al., 2009 call surrounding sediments with absence of any macroscopically observed carbonates. Therefore, the ratio of adsorbed and extractable methane to bulk methane in even non-seep environment is poorly understood and lacks firm quantifications due to substantial analytical errors. In our study area where free gas escapes the seabed, the contribution of occasional desorption of gas strongly-bound to clay minerals is deemed to be negligible compared to abundance of its more labile forms (dissolved and free gas).

The issue of mixing free gas and dissolved gas in headspace samples is inevitable and, we believe, has become a condition well known in marine geoscience. Headspace gas analysis shows concentration of only dissolved gas if this concertation is lower than the solubility limit under P and T conditions of a laboratory where the samples are collected. Concentrations measured in our shallow sediments are below this critical value, thus representing dissolved gas only.

Analyses of pore water collected with rhizons are not optimal due to long exposure (at least an hour) of water drops to the air in the syringes that always have some dead volume. During such sampling dissolved gas gets equilibrated with atmospheric gas causing loss of methane in analyte that is hard to trace and account for.

Despite this, even if our method did result in some amount of free gas and sorbed gas getting included in our measurements, the same method was used for all the pingos and the point was to compare between them. Therefore, even if our method overestimates dissolved methane concentrations, it does so equally for pingo 5 and the other pingos. And our measurements, which were lower at pingo 5 compared to the other pingos, likely indicates that dissolved methane is also lower at pingo 5.

Sulphide

Dissolved reduced sulphur species are utilised by chemoautotrophic free-living and symbiotic bacteria as an energy source. The concentration of dissolved "sulphide" (H2S + HS- + S=) and thiosulphate is thus an important measurement. The authors state that sulphide was below the detection limit in the bottom water and in the upper "few" cm of the sediment of all the cores. However, sulphide must have been present to support the bacterial mats visible on the surface. The detection limit is not given but since the lowest standard used in the assay was 40 μM (Hong et al., 2017) the method described may not have been able to detect concentrations of a few μM. Many thiotrophic symbiotic associations exist in sediments with dissolved sulphide concentrations of < 1 μM. It is very difficult to prevent oxidation of low concentrations of sulphide in pore water and since the samples were not analysed immediately, it is probable that oxidation occurred during preservation and storage. H2S would have been carried into the upper sediment and the water column in the methane bubbles as well as in the associated water plume (Reeburgh 2009, Dando et al. 1994a). In addition, the drawdown of seawater induced by the rising methane bubbles (O'Hara et al. 1995, Zimmermann et al. 1997) would have locally generated reduced sulphur species from iron sulphides within the sediment (Dando et al. 1994b) as well as producing a halo of less reducing areas surrounding the bubble outlets.

Response: Dr. Dando is correct, the detection limit was 40 μM and we agree that thiotrophic symbioses can exist with much lower concentrations of dissolved sulphide.

We also agree that the presence of bacterial mats indicate the presence of dissolved sulphide at the water-sediment interface (although the mats are very patchy and usually quite small). We can therefore remove the sentence on sulfide toxicity with respect to non-seep fauna.

However, the main points we were trying to make were with respect to comparing sulphide concentrations between the different pingos and they are: 1) sulphide concentrations likely do not differ between the pingos and 2) we might have been unable to detect sulphide in the bottom water, but that is the case for both pingo 5 and the other pingos. We believe that these points are still valid. However, we can emphasize our inability to detect low concentrations of sulphide, both within the sediment and in the bottom water, but doing this would not alter the conclusion we came to, that 'GHP5 is not deficient in this regard (sulphide) either, in comparison to the other pingos'. We also would like to emphasize that, even though we did not measure sulphide concentration immediately, the porewater samples were collected by rhizons which has been shown to prevent oxidation (Seeberg-Elverfeldt Jens et al., 2005) of the samples. We also preserved the samples by adding saturated Ac(OAc)2 < 30 minutes after the rhizons were disconnected from the syringes. Such preservation measure is widely used in the literature and can prevent the oxidation of sulphide.

With respect to the downward seawater irrigation due to rising bubbles in the sediments, Hong et al. (2017) have shown that it is unlikely to occur. Briefly, if seawater indeed intruded from the bottom seawater to the surficial sediments, one would expect seawater concentrations for all porewater species. Hong et al. (2017) have shown that this is not the case from the 100 fold higher ammonium detected in the first 5-10 cm of the sediments. We can therefore confidently exclude the impact on sulfide concentration by such so-called bubble irrigation mechanism.

Identification of biota

The statements "Visible fauna (at least a few centimeters across) were identified" (p5) and the statement in the following paragraph that "Numerous individuals of siboglinid worms were seen", appear contradictory. *Oligobrachia haakonmosbiensis* is large for a frenulate, with a tube diameter of 0.6-0.9 mm (Smirnov, 2014). It would be useful to have a high resolution image, perhaps as a supplementary file, to show how these individual siboglinids were visible in the photographs, since a lot of the Discussion is based on their presence or absence.

Response: We can add a supplementary figure, to show what the worms look like in the images. It should be kept in mind that individuals of *O. haakonmosbiensis* were not seen or marked. Clumps or mats of them were seen and these aggregations were outlined (Methods, pages 5-6). Since this point was not clear, we can change this portion of the Methods section to clarify this.

Most of the identifications relied on interpretations of images taken by a towed camera with a resolution of 16 million pixels and with stereo cameras mounted on a ROV with a resolution of 1.4 million pixels. We are not told the respective field of views photographed by these cameras so it is not possible to estimate the respective resolutions. It would help interpretation if the authors would calculate the sizes that the respective pixels represent. Rough calculations, from the dimensions given in Figure 2, suggest that the pixel size in images from

the stereo cameras may have been inadequate to resolve smaller organisms, such as Oligobrachia tubes, 0.6-0.9 mm in diameter (Smirnov, 2014) unless they occurred in clumps.

Response: As stated above, the worms did appear in clumps and these clumps were outlined as polygons in a GIS as opposed to every individual being marked. It is absolutely true that single individuals would not be visible in the images. Figure 2 does have scales in the mosaics/transects, but we can add scales in the individual images as well to give a better idea of the fields of view.

The core samples at GHP 5 were taken around the periphery of the pingo (Figure 1) so that it is not possible to deduce from these that *Oligobrachia* was absent from pingo 5.

Response: The deduction that *Oligobrachia* was absent from pingo 5 is not based only on the core samples. This deduction is based on the images and mosaics from the site. It is also based on our extensive surveys of the pingo before imaging for mosaicking purposes was conducted. Therefore, we did not conclude that *Oligobrachia* was absent from pingo 5 just because we did not recover them in the cores (and 1069 at least is not from the periphery). We concluded that they were absent because we spent a considerable amount of time surveying the pingo with the ROV's HD video camera, and did not see the clumps that are so abundant on the other pingos. We are quite confident that the worms are more or less absent at pingo 5. It is certainly possible that a few, scattered individuals are present and these solitary individuals would not be visible in any kind of imagery. However, this is one of the limitations that always exist with image-based studies and is the standard shortcoming that has to be taken into consideration when using an image based approach. We can add this caveat to the discussion. Nonetheless, mats of *Oligobrachia* are not present at pingo 5, and this is an important difference between pingo 5 and the other pingos, and the fundamental statement that we address in this manuscript.

Another problem in comparing tow cam and ROV pictures is that the ROV imaging was always from fairly discrete areas on pingo 5 while the tow cams were transects covering from the outside into the centre of the pingo (Figure 1). This might explain why Nothria, for example, was identified in all the tow cams but not in the ROV pictures. The tow cam epifaunal data, presented in Table 1, should therefore be divided into "on pingo" and "off pingo" sections. The reason that TC25 GHP3 clusters with the GHP5 ROV camera tracks is probably because tow cam 25 has the greatest proportion of offpingo track of any of the tow-cams.

Response: It is true that the tow cam images are transects whereas the ROV images cover more discrete areas (although ROV mosaic 3 is more a series of transects than a mosaic). That is why we  chose  to neither discuss the overall community structure, nor to make comparisons of overall community between pingo 5 and the other pingos, except very briefly, and to mention the diversity indices. We included a figure of the results of the community analyses in order for these results to be available for everyone, but we refrained from discussing these results, other than very briefly because we agree that the different ways in which the pingos were imaged could be a factor that muddles the overall community characteristics. This is why we instead chose to focus on the presence or absence of *Oligobrachia* since this is a trend that we feel confident of comparing between the different pingos, as stated above.

It is more likely that the absence of Nothria from the ROV images is due to the imaging capabilities, and not because the ROV images were discrete mosaics. This is because we could not see Nothria in ROV images from pingo 3, i.e., the ROV images over pingo 3 that could not be used because the navigation data was not useable. If the absence of Nothria from pingo 5 was simply due to the locations in which the images were taken, then they would be visible in the ROV images taken over pingo 3. Their absence from these images suggests that they are not visible in the ROV camera. Regardless, however, Nothria was not used in the community analysis, nor in comparisons between the pingos.

All of the images and faunal data from them is more or less 'on-pingo.' We did not use any 'off-pingo' images for our quantitative analyses, therefore the data presented in Table 1 cannot be divided into these two categories. There is no clear boundary that distinguishes the seafloor as being part of a pingo versus not. Therefore, we only used images that appear to be part of the different pingos. We know that our navigation data was not perfect for this, but we also looked at the images and the presence of obvious signs of seepage (carbonate crusts, bacterial mats, worm tufts, etc.) as evidence of the images being taken from the seeping pingos. It is entirely possible that some images included in our study are in fact, slightly off the pingo in question, but this error would exist for all the pingos from which the tow cam images were taken. It is true that this error would not occur for the ROV images over pingo 5 but, as stated above, that is why we did not go into any detailed discussions about comparing overall communities.

The only time that we did consider 'off-pingo' images was in a transect to the west of pingo 3 (at least 1 km away), which we discuss only qualitatively, where we introduce the idea that seepage areas might have higher species diversity than background, non seep affected seafloor.

In short, all the tow cam images are, to the best of our ability, with the given constraints of the study, taken over the pingos themselves. It would therefore be inappropriate to classify any of them as being 'off-pingo.' But we agree that off-pingo areas could have been included in the tow cam images, which makes comparisons with the discrete mosaics on pingo 5 difficult. However, we acknowledge this shortcoming and as a result, do not discuss community differences between pingo 5 and the other pingos. We stick only to the main difference, ie, the presence or absence of frenulates, which we do feel confident is a real difference between pingo 5 compared to the others.

The frenulates observed were identified from specimens in the core samples and density estimates for them were calculated from the densities observed in the cores. It would have helped the interpretations if information had been provided on the depth they reached in the sediment. This may be site specific, since the penetration depth of a species has been shown to vary between cores (Dando et al. 2008). *Oligobrachia haakonmosbiensis* were reported penetrating the sediment to a depth of 55 cm at the Hîkon Mosby mud volcano (Lösekann et al. 2007).

Response: We did not include this because we did not conduct any good, exhaustive measurements on how far the tubes penetrated (and we had different types of cores,

which all affect the animals differently when they are retrieved). Roughly, we can say that the tubes reached 50-60 into the sediment, which is in the same range as what was seen at the Håkon Mosby mud volcano by Lösekann et al., (2008) and Gebruk et al. (2003). We can add this information to the text.

It would also be helpful to know whether other macrofauna were recovered from the cores, since most faunal species with chemoautotrophic bacteria at shallower seeps are infaunal and would not show on surface photographs. An example is at a methane seep at 170 m depth in the N. Sea where 3 such species were found living within the sediment and shells of a fourth, the bivalve *Lucinoma borealis*, were recovered (Dando et al. 1991, Dando 2001): noepifauna with chemoautotrophic symbionts were observed. Many frenulates have tubes completely buried within the sediment; thus chemosynthesis is probably more common at the pingo sites than this study of mainly epifauna suggests.

Response: We agree that there might be infaunal species that are chemosymbiotic, and we mention this in the text. We also tried to emphasize that we are only considering animals that are visible in images in this study. However, this might not have always been clear, especially when we talk about *Oligobrachia* being the only confirmed chemosymbiotic species. We can change the text to make sure that every time we talk about a trend like that, we specify that we are referring only to larger animals, visible in images and that smaller, infaunal animals are not taken into account in this study.

Discussion
On discussing the distribution of the frenulate *Oligobrachia*, the authors wrote: "the image transects containing siboglinid frenulates were much less even in terms of species abundances compared to the transect and mosaics which did not contain any frenulates". This would be expected since it has been shown that, on the Rockall slope, frenulate distribution did not cluster with most other taxa and there was an inverse relationship between frenulate density and the density of other benthos (Dando et al. 2008). This was explained because sediment disturbance by other organisms would increase sulphide oxidation and displace, or bury, the thin tubes of the frenulates. It should be noted that in the latter study most of the frenulates had a low abundance and none of the tubes projected from the sediment, if they did at all, as far as those of *Oligobrachia haakonmosbiensis* and thus would not provide an epifaunal habitat. In the one obligate, methane seep frenulate species that occurred in high densities, *Siboglinum poseidoni* (Dando et al. 1994c), no epifauna were noted between or above the projecting tubes.

The authors consider that chemoautotrophic primary production at the pingos might exceed the photosynthetic primary production reaching the sea floor. The examples they cite are from deeper water, where less photosynthetic production reaches the sea floor. This is unlikely to be true at 400 m where much more photosynthetic production will reach the seabed. As shown in a comparative study (Bernadino et al. 2012), the isotopic difference between background and seep fauna was much lower at the Eel River seeps (250-500m) than at deeper seeps at 770 m depth and deeper. Isotopic evidence of food inputs is needed to support the authors' hypothesis. Since methane solubility increases with pressure there will, potentially, also be more methane available to the biota at deeper sites.

Response: We do have isotopic evidence of chemosynthetic food input at the pingo site, but these results are part of a separate article and we are not really at liberty to discuss them yet. Our goal was not to show that chemoautotrophic production exceeds photosynthetic production per se, we just wanted to include the notion that local primary production, in the form of chemoautotrophy occurs at the site. Instead of saying that 'autochthonous chemoautotrophic primary production tends to exceed photosynthesis derived detrital food supply (page 12), we can change the text to say that both photosynthetically derived and chemosynthesis based organic matter is likely available at the pingos. The point here was to try to explain why certain animals were seen to appear to show a preference for seep based habitats such as the frenulates, carbonates or bacterial mats. With respect to nutrition, we mentioned that some animals might be grazing on bacterial mats, which could in turn affect higher order consumers and explain their distribution among bacterial mats. Therefore, changing the text so that it does not imply that chemoautotrophic input exceeds photosynthetic input will not change the overall message and so we can easily modify the text to avoid this confusion.

Regarding sulphide in the upper sediment, p.14 line 11: only 1 measurement at 5 cm depth is shown in Fig. 4 (for core 1045, off the edge of pingo 3). The exact value for this sample is not shown. However, a single measurement does not justify the statement that "sulfide was not detectable - - - even in the upper 5 cm of the sediment at the pingo site". Should this read "sites"?

Response: The value is 0. The 0 values for the other cores should also be shown in this figure and we apologize that they are not there. Even at 10 cm, we often could not detect sulphide in the cores, which is the reason for the statement on page 14, line 11, that sulphide was not detectable in the bottom water or even in the first 5 cms. That is, this statement was not based on a single measurement from one core at all, but we agree that the 0 values for the other cores should be shown on this figure and we will rectify that.

As mentioned above, it is probable that sulphide was present at significant concentrations for the biota, including the bacterial mats at the surface, but was not detected using the stated analytical procedure. Serov et al. (2017 Fig. S2C) shows a picture from one of the pingos with a white bacteria mat, presumably of sulphur-oxidising bacteria, on top of "tubeworms" that project approximately 4 cm above the sediment, if the scale on the photograph is correct. If these are sulphur-oxidising bacteria then sulphide or thiosulphate must be present in the water column. The "tubeworms" are approximately 10 mm across, measured against the scale on the photograph, and thus cannot be Oligobrachia.

Response: As acknowledged above, we will change the text so that we clarify that we did not detect sulphide, but there could be concentrations lower than what we detected, including using bacterial mats as evidence for this.

With respect to the image in Serov et al., based on previous descriptions of *Oligobrachia,* it would appear that the worms with filamentous bacteria on them are too large to be *Oligobrachia*. However, we have sampled these worms and they are, in fact, *Oligobrachia* (manuscript in preparation). The bacteria can form large, dense colonies on the tubes of the worms, so that they appear much larger than the tubes themselves. Below is an image that shows this. This was also seen in nearby sites such

as in Storfjordrenna (Åström et al., 2016) and a site of pingo-crater complexes in Bjronøyrenna (publication in preparation).

[Figure]

P14, line 21 and subsequent text: "this particular image transect did not contain frenulates." *Oligobrachia haakonmosbiensis* havs large tubes for a frenulate, many species have tubes 100 μM or less across and would not be visible if they did project above the surface, although many species are completely buried within the sediment. More than one species frequently occur in the same core sample (Dando et al. 2008) so that it is not possible to state that frenulates were absent. To be correct you should write that "this transect did not show any 'visible' frenulates".

Response: We can change this as suggested. As we mentioned before, we were trying to emphasize that only animals visible in images are considered in this study, but obviously, we have not emphasized this enough and we will make sure that this point is very clear.

GHP5 gas release: in many submarine seeps gas is only released at low tide when slight differences in bottom pressure cause the sub-surface gas volume to increase. At other frequently visited methane seep sites, such as the Scanner pockmark, continuous gas release may, or may not, be present during any given cruise. In the absence of data on the tidal conditions when observations were made over GHP5 it is not possible to state that gas was not released from this pingo.

Response: It is indeed true that gas seepage can be induced by tidal effects. We however do not think this can explain the contrast in gas flare activity between pingo 5 and the other pingos as tidal changes should pose the influence on all pingos that have almost the same water depth and are within an area of only about 2km$^2$. Gas flare data was acquired in 2016 together with the 3D seismic survey, and all in all, the survey took two days. During this time-span, we detected no gas flares from GHP5, and thus we can rule out a potential tide-controlled leakage. Furthermore, as is mentioned in the manuscript, several cruises were conducted, over three years and during different seasons, all of which consistently documented the absence of any flare activity over pingo 5.

The enhanced reflectors below GHP5 indicate subsurface gas and, on enlargement of Figure 6, it is possible to see a small gas "chimney" towards the edge of the pingo (see Fig 6 section), although this is considerably smaller than the chimneys below the other pingos. Core 920 on the edge of GHP 5 contains methane of thermogenic origin (Serov 2017, Table S1), implying a deep source for the gas.

Response: The enhanced reflectors below GHP5 may indicate pockets of subsurface gas, buried carbonates or gas hydrate (but not necessarily active gas migration feeding the GHP). The seismic data show lower amplitude dipping reflectors underneath the unconformity below GHP5. Active seismic chimneys are normally represented by distinct acoustically masked areas (such as under pingos 1-3) or a vertical pipe structure often accompanied by velocity-related anomalies, which we don't observe below GHP5. A narrow zone of weak acoustic blanking under the margin of pingo 5 may indicate either very low to negligible fluid migration, or a fault zone. On the neighboring inlines and crosslines in our seismic volume, this feature appears even less prominent; therefore, we conclude that there is no significant fluid/gas migration underneath GHP5. On the other hand, underneath the other pingos, prominent seismic chimneys occur. We can change the text so that instead of writing "lack of chimney.." in the manuscript we can write "no prominent seismic chimney" underneath GHP5.

Active release of methane from the sediment will channel the methane into the higher porosity release channels. The sediment at the sides of these channels will have a low methane concentration, due to the down-flow of seawater from the sediment surface (Dando et al. 1994a, O'Hara et al. 1995). Thus it is not correct to argue that methane release will stimulate overall sulphate reduction and methanotrophy in a seeping pingo when comparedto a non-seeping pingo with a high sediment methane concentration (p17 first paragraph). Microbes may also be removed from the system by the rising fluids.

Response: As our reply for the previous comments and the results from Hong et al. (2017), from the concentration of ammonium in the porewater, we can confidently exclude the possibility of downward seawater flow into the sediments. We therefore do not think such argument is relevant in our study sites.

Furthermore, sulfate reduction rates were measured independently of methane. Therefore, our argument that sulfate reduction rates are different at pingo 5 is still valid. Overall, the point is that pingo 5 is different from the other pingos. There is lower methane concentration in the sediment (whether this is strictly dissolved methane or not), there were no hydrates recovered from pingo 5, there are no prominent seismic chimneys below pingo 5, sulfate flux rates were lower at pingo 5, and no gas flares rising into the water column were seen at pingo 5. Together, these seem to indicate that the geochemical conditions at pingo 5 are different, which could account for the absence of frenulates. We even have preliminary results indicating that the microbial community at pingo 5 is different, including that ANMEs make up less of the total bacterial/archaeal community at pingo 5 (Klasek et al., in prep, which is referred to in the manuscript), which further supports our argument.

In short, we believe that our overall conclusion, of different geochemical conditions at pingo 5, compared to the other pingos, is nonetheless valid, and could account for the absence of frenulates from pingo 5.

Sulphate reduction (p16): "In most cores, the ratio of inorganic carbon to sulfate consumption was found to be close to 1:1 regardless of depth (both GHP5 and GHP3). The one exception was core 1048 from GHP5, for which, almost all values were closer to the 2:1 ratio." Core 1048 is shown in Figure 1 to be the furthest away from any pingo, i.e. it is in background sediment. Thus it should be no surprise that in this core sulphate reduction is not dependent upon the presence of methane.

Response: This is correct.

"The dual need for inorganic and organic carbon sources (plus thiotrophic chemoautotrophy)likely results in frenulates **in general**, and, *O. haakonmosbiensis* specifically, relying heavily on a highly active sediment methanotrophic microbial community"(p16 line 31). This is not true as a general statement. Dando et al. (2011), in a study of the relationship between 10 species of frenulates and the chemistry of their habitat, found that, with the exception of 1 obligate methane seep species, all occupied sediments where the dissolved methane concentration was < 1 μM.

Response: We did not phrase this correctly. The idea was to introduce a hypothesis, that *O. haakonmosbiensis* relies on an active microbial community because they might be obtaining their nutrition from both their symbionts and the surrounding sediment. We can rephrase this.

"On the other hand, at GHP5, seepage of methane is low due to the lack of a sub-surface gas chimney. Methane is still present in the sediment, but in lower concentrations and as a result, methanotrophic microbes are likely less abundant and methanotrophic activity is considerably lower, as evidenced by lower AOM linked sulfate flux rates" p17. As mentioned earlier the authors do not know the concentration of available methane in the sediment and hence cannot make such comparisons regarding different methane concentrations at different sites. The values in Figure 4 may just equate to the amount of authigenic carbonate in the samples. A small gas chimney appears to be visible below GHP5 in Figure 6.

Response: As stated above, we disagree that we did not measure dissolved methane. Additionally, we must emphasize that the presence of gas hydrates was observed in most of the sediment cores recovered from pingos 1-3, while no gas hydrate was recovered from any of the sediment cores in pingo 5. Dissolved methane concentration must be high enough (i.e., at saturation level) to allow for the presence of gas hydrates, which is the case for most the sediment cores, but not pingo 5. This is quite solid evidence to support our inference and therefore contrasting methane concentrations between pingo 5 and the other pingos.

We do not understand the rationale behind "methane concentration may just equate the amount of authigenic carbonates in the sediments". Precipitation of authigenic carbonates depends on the saturation state of carbonate minerals, which is a function of the availability of DIC, calcium, and magnesium in the porewater. The supply of calcium and magnesium in the porewater is independent of the supply of methane in the sediments. Of course a faster turnover of methane through AOM can accelerate the precipitation of authigenic carbonate precipitation but there is no sign showing the absolute amount of methane and authigenic carbonate should be in any way be related.

As discussed above, despite there being some blanking under pingo 5, the seismic data indicates that there are pockets of gas, but not necessarily active gas migration. Furthermore, no hydrates were recovered from pingo 5, which also suggests less seepage there. Combined, we believe that this suggests that dissolved methane concentrations are lower in the sediment at pingo 5, irrespective of whether one is convinced we measured dissolved methane in our samples or not. In any case, sulfate flux rates were measured independently of methane, and they suggest lower methanotrophic activity at pingo 5, which is the main crux of our argument.

The discussion regarding hydrothermal vents is not very appropriate for this paper with respect to O. *haakonmosbiensis*. This is a cold-water species that penetrates approximately 0.5 m into the sediment. At vent sites the temperature within the sediment would, almost certainly, be lethal for the species.

Response: We were referring to the 'lower temperature' zone of seeps where *Sclerolinum* is found, but we agree, we can remove the discussion related to hydrothermal vents.

Although *O. haakonmosbiensis* was the only metazoan with chemoautotrophic symbionts found, it does not mean that it was the only one present, since the infauna, where, for example, other frenulates and thyasirid and lucinid bivalves might be expected, was not studied. It is therefore also not correct to state that "the community at the pingos does not contain specialized seep endemics" (p22 line 13) since the infauna were not studied and *O. haakonmosbiensis*, if distinct from *O. webbi*, is probably a seep obligate species. "Endemic" is incorrect in this context since it refers to geographic regions, not habitats.

Response: We can change to say seep specific or seep obligate. And yes, we agree that there might be infaunal community members that are seep obligate, and we do mention this in the text (thyasirids). Once again, we will have to make sure that we clearly state that we are talking about larger, visible fauna. We acknowledge that the frenulates at the site might be seep obligate (e.g., page 19, page 12). But we can also change any discussions about seep obligates in the overall community and make sure that we do not say that they are absent or completely lacking, but rather, that only one species has so far been seen (and again, make sure that the scale we are referring to is large  animals visible in images).

Figure 1
It would aid interpretation if the positions of the observed gas flares were pinpointed in figures b-e.

Response: We can add these.

Figure 2, 8 & 9
These would benefit from scales in the camera pictures, since the laser spots, when present, are difficult to see.

Response: We can add them.

Figure 3

Figure 3b has TC21 and TC18GHP3 plotted on top of each other, including the labels, so it is not clear what this point represents.

> Response: This is because they are so similar, that they end up being right on top of each other. We can include an explanation in the figure caption.

Figure 4
The lines after the final points appear to be extrapolated randomly. If this is because the graphs are part of larger ones and have been cut off at 60 cm then it would be sensible to give the depth and values of the next points in parenthesis at the end of the lines. The coloured open circles are not well differentiated at the scale of the Figure and should be replaced by coloured filled circles to differentiate the cores.

> Response: We can make these changes.

Figure 6
I think the vertical scale is m depth below the sea surface and not sediment depth. Fig, 6b is too small to be useful without enlargement.

> Response:We can make 6b larger. Yes, the vertical scale should be meters below sea level and we can change this.

Discussion
The term "megafaunal taxa" is used in the Discussion. Megafauna are large animals such as cetaceans and large fish. The correct term for the taxa observed is "macrofauna"

> Response: The distinction between megafauna and macrofauna is somewhat subjective and different people have different opinions on how to use the two terms. We use megafauna for this manuscript since we refer to animals large enough to be seen easily with the naked eye. We consider macrofauna to be smaller animals that are retained on a 0.3 mm or 0.5 mm sieve (this cut off seems to vary between studies) and are not easy to see through imagery. This definition is certainly subjective as well, but it is in accordance with many other similar seep and vent studies and we chose to use this terminology to be consistent with other studies with similar methodologies (Amon et al., 2017; Baco et al., 2010; Bowden et al., 2013; Hessler et al., 1988; Lessard-Pilon et al., 2010; Marcon et al., 2014; Podowski et al., 2009, 2010; Rybakova (Goroslavskaya) et al., 2013; Sellanes et al., 2008).

P14 lines 20 & 21, Fig. 2 should read Fig. 3

> Response: Thanks for pointing this out, we can change this.

**References**
Dando, P. R., Austen, M. C., Burke, R. J., Kendall, M. A., Kennicutt, M. C., Judd, A. G., et al. (1991). Ecology of a North Sea pockmark with an active methane seep. Marine Ecology Progress Series, 70: 49- 63.

Dando, P. R., Jensen, P., O'Hara, S. C. M., Niven, S. J., Schmaljohann, R., Schuster, U., et al. (1994a). The effects of methane seepage at an intertidal / shallow subtidal site on the shore of the Kattegat, Vendsyssel, Denmark. Bulletin of the Geological Society of Denmark, 41: 65-79

Dando, P. R., Ridgway, S. A., & Spiro, B. (1994b). Sulphide 'mining' by lucinid bivalve molluscs: demonstrated by stable sulphur isotope measurements and experimental models. Marine Ecology Proress Series, 107: 169-175.

Dando, P. R., Bussmann, I., Niven, S. J., O'Hara, S. C. M., Schmaljohann, R., & Taylor, L. J. (1994). A methane seep area in the Skagerrak, the habitat of the Pogonophore, Siboglinum poseidoni, and the bivalve mollusc Thyasira sarsi. Marine Ecology Progress Series, 107: 157-167

Dando, P. R. (2001). A review of pockmarks in the UK part of the North Sea, with particular respect to their biology. Technical Report TR 001 Strategic Environmental Assessment ⊩ SEA2. London: Department of Trade and Industry.

Dando, P. R., Southward, A. J., Southward, E. C., Lamont, P., & Harvey, R. (2008). Interactions between sediment chemistry and frenulate pogonophores (Annelida) in the north-east Atlantic. Deep- Sea Research I, 55: 966-996

Ertefai, T. F, Heuer, V B, Prieto-Mollar, X., Vogt, C., Sylva, S. P, Seewald, J. S., Hinrichs, K-U. (2010) The biogeochemistry of sorbed methane in marine sediments. Geochimica et Cosmochimica Acta, 74: 6033-6048

Hong, W.-L., Torres, M. E., Carroll, J., CrÄmiÅre, A., Panieri, G., Yao, H. and Serov, P. (2017) Seepage from an arctic shallow marine gas hydrate reservoir is insensitive to momentary ocean warming, Nature Communications, 8, ncomms15745, doi:10.1038/ncomms15745

Ijir, A. , Tsunogai, U. Gamo, T., Nakagawa, F., Sakamoto, T., Saito, S. (2009) Enrichment of adsorbed methane in authigenic carbonate concretions of the Japan Trench. Geo-Marine Letters, 29:301⊩308

Lösekann, T., Knittel, K., Nadalig, T., Fuchs, B., Niemann, H., Boetius, A. and Amann, R. (2007) Diversity and Abundance of aerobic and anaerobic methane oxidizers at the Haakon Mosby Mud Volcano, Barents Sea, Applied and Environmental Microbiology, 73: 3348⊩3362

Reeburgh W. S. (1969) Observations of gases in Chesapeake Bay sediments. Limnology and Oceanography, 14: 368-375.

Serov, P., Vadakkepuliyambatta, S., Mienert, J., Patton, H., Portnov, A., Silyakova, A., Panieri, G., Carroll, M. L., Carroll, J., Andreassen, K. and Hubbard, A. (2017). Postglacial response of Arctic Ocean gas hydrates to climatic amelioration. Proceedings of the National Academy of Sciences, 114: 6215⊩6220

Smirnov, R. V. (2014). A revision of the Oligobrachiidae (Annelida: Pogonophora), with notes on the morphology and distribution of Oligobrachia haakonmosbiensis Smirnov. Marine Biology Research, 10: 972-982 Zimmermann, S., Hughes, R. G., & Flugel, H. J. (1997).

The effect of methane seepage on the spatial distribution of oxygen and dissolved sulphide within a muddy sediment. Marine Geology, 137: 149-157

Figure 6 section

[Figure]

approximate line
of proposed gas
chimney

References

Amon, D. J., Gobin, J., Dover, V., L, C., Levin, L. A., Marsh, L., et al. (2017). Characterization of Methane-Seep Communities in a Deep-Sea Area Designated for Oil and Natural Gas Exploitation Off Trinidad and Tobago. *Front. Mar. Sci.* 4. doi:10.3389/fmars.2017.00342.

Åström, E. K. L., Carroll, M. L., Jr, W. G. A., and Carroll, J. (2016). Arctic cold seeps in marine methane hydrate environments: impacts on shelf macrobenthic community structure offshore Svalbard. *Mar. Ecol. Prog. Ser.* 552, 1–18. doi:10.3354/meps11773.

Baco, A. R., Rowden, A. A., Levin, L. A., Smith, C. R., and Bowden, D. A. (2010). Initial characterization of cold seep faunal communities on the New Zealand Hikurangi margin. *Mar. Geol.* 272, 251–259. doi:10.1016/j.margeo.2009.06.015.

Bowden, D. A., Rowden, A. A., Thurber, A. R., Baco, A. R., Levin, L. A., and Smith, C. R. (2013). Cold seep epifaunal communities on the Hikurangi margin, New Zealand: composition, succession, and vulnerability to human activities. *PloS One* 8, e76869. doi:10.1371/journal.pone.0076869.

Gebruk, A. V., Krylova, E. M., Lein, A. Y., Vinogradov, G. M., Anderson, E., Pimenov, N. V., et al. (2003). Methane seep community of the Håkon Mosby mud volcano (the Norwegian Sea): composition and trophic aspects. *Sarsia* 88, 394–403. doi:10.1080/00364820310003190.

Hessler, R. R., Smithey, W. M., Boudrias, M. A., Keller, C. H., Lutz, R. A., and Childress, J. J. (1988). Temporal change in megafauna at the Rose Garden hydrothermal vent

(Galapagos Rift; Eastern tropical Pacific). *Deep Sea Res. Part Oceanogr. Res. Pap.* 35, 1681–1709.

Hong, W.-L., Torres, M. E., Carroll, J., Crémière, A., Panieri, G., Yao, H., et al. (2017). Seepage from an arctic shallow marine gas hydrate reservoir is insensitive to momentary ocean warming. *Nat. Commun.* 8, ncomms15745. doi:10.1038/ncomms15745.

Lessard-Pilon, S., Porter, M. D., Cordes, E. E., MacDonald, I., and Fisher, C. R. (2010). Community composition and temporal change at deep Gulf of Mexico cold seeps. *Deep Sea Res. Part II Top. Stud. Oceanogr.* 57, 1891–1903. doi:10.1016/j.dsr2.2010.05.012.

Lösekann, T., Robador, A., Niemann, H., Knittel, K., Boetius, A., and Dubilier, N. (2008). Endosymbioses between bacteria and deep-sea siboglinid tubeworms from an Arctic Cold Seep (Haakon Mosby Mud Volcano, Barents Sea). *Environ. Microbiol.* 10, 3237–3254. doi:10.1111/j.1462-2920.2008.01712.x.

Marcon, Y., Sahling, H., Allais, A.-G., Bohrmann, G., and Olu, K. (2014). Distribution and temporal variation of mega-fauna at the Regab pockmark (Northern Congo Fan), based on a comparison of videomosaics and geographic information systems analyses. *Mar. Ecol.* 35, 77–95. doi:10.1111/maec.12056.

Podowski, E. L., Ma, S., Luther III, G. W., Wardrop, D., and Fisher, C. R. (2010). Biotic and abiotic factors affecting distributions of megafauna in diffuse flow on andesite and basalt along the Eastern Lau Spreading Center, Tonga. *Mar. Ecol. Prog. Ser.* 418, 25–45.

Podowski, E. L., Moore, T. S., Zelnio, K. A., Luther, G. W., and Fisher, C. R. (2009). Distribution of diffuse flow megafauna in two sites on the Eastern Lau Spreading Center, Tonga. *Deep Sea Res. Part Oceanogr. Res. Pap.* 56, 2041–2056.

Rybakova (Goroslavskaya), E., Galkin, S., Bergmann, M., Soltwedel, T., and Gebruk, A. (2013). Density and distribution of megafauna at the Håkon Mosby mud volcano (the Barents Sea) based on image analysis. *Biogeosciences* 10, 3359–3374. doi:10.5194/bg-10-3359-2013.

Seeberg-Elverfeldt Jens, Schlüter Michael, Feseker Tomas, and Kölling Martin (2005). Rhizon sampling of porewaters near the sediment-water interface of aquatic systems. *Limnol. Oceanogr. Methods* 3, 361–371. doi:10.4319/lom.2005.3.361.

Sellanes, J., Quiroga, E., and Neira, C. (2008). Megafauna community structure and trophic relationships at the recently discovered Concepción Methane Seep Area, Chile, ~36°S. *ICES J. Mar. Sci.* 65, 1102–1111. doi:10.1093/icesjms/fsn099.

---

## Short Comment (SC1) · 9 Apr 2018

Reply to Interactive comment II on
Relating the geochemistry to the species distribution of the macrofauna at seeps is always very difficult. Concentrations of possible energy sources, both in the surface layers, and within the sediment depth to which most macrofauna occur, can change by

an order of magnitude or more over a distance of 10-50 cm from the seep outlet. In this study the positions of the cores with respect to the camera tracks are not that well constrained. Core 920 is shown here as NE of GHP5 while in Hong et al. (2017) it is shown as SE of GHP5.

Response: We agree completely, but we urge readers to remember that constraining the biology based on the geochemistry at a fine scale was not the aim of this study. We were instead looking at more general patterns over a larger scale. Namely, that pingo 5 does not have extensive mats of siboglinids like the other three pingos. We initially thought that lower sediment concentrations would explain this trend, but as it turns out, pingo 5 does not have lower sediment sulfide concentrations compared to the other pingos. However, there are multiple lines of evidence that suggest that overall geochemical conditions at pingo 5 are different, and we believe that these could explain the absence of siboglinids from pingo 5. These differences are: 1) lower methane concentrations at pingo 5 (even if it is not strictly only dissolved methane), 2) no hydrates were recovered from pingo 5 but were from the other pingos, 3) there are no prominent seismic chimneys below pingo 5, 4) sulfate flux rates are lower at pingo 5 compared to the others, 5) there are no rising gas flares into the water column from pingo 5, but there are from all the other pingos, and 6) ANMEs make up less of the total microbial community at pingo 5 compared to the others. Together, we believe that these differences suggest that overall at pingo 5, there is likely to be lower methane flux and a less active methanotrophic microbial community (i.e. lower AOM rates).

We checked our coordinates and it appears that there is a mistake in Hong et al., 2017.

We should mention that cores were not taken in sync with the imaging efforts. We are well aware that the cores do not line up with the mosaics or transects. And therefore, fine scale comparisons of geochemistry with biology is not possible. But we do believe that at the scale of one pingo as a whole compared to another, our sampling efforts were sufficient. In fact, the similar sulfide profiles from cores taken at pingo 5 and the other pingos indicate that our sampling efforts were adequate to obtain a general
overview of geochemical conditions at the scale of individual pingos. Had we only got 'peripheral' data from pingo 5, then the sulfide concentrations from pingo 5 would be consistently lower, but this was not the case (instead, it was very similar to the other pingos).

My earlier comment regarding false dissolved methane readings by adding equal volumes of 1 M sodium hydroxide to samples for the analysis of free methane was influenced by my own early studies in which samples were not analysed at sea and had a longer storage time in contact with NaOH. I accept that a short exposure may not bias the results to the extent I believed, although it would be good to have results from an untreated pore water sample for comparison. Indeed Ertefai et al. also used 1 M NaOH treatment before measuring free methane and then continued the treatment for a longer period to measure adsorbed methane. Timing is everything!

Response: We agree with Dr. Dando that timing is crucial. In the future research we will conduct test measurements with and without NaOH solution in samples at different times after sampling. It is known that results of FID GC measurements of headspace samples require interpretation. We interpret measured concentrations from the samples collected in extremely active seepage site with massive gas hydrate layers within 1-3 m of sediment column, bacterial mats on the seabed and authigenic formations on the seabed as concentrations of the labile (dissolved) methane. GHP 5 clearly shows indications of some gas seepage (mats, fauna, etc.) making us confident that dissolved gas is present in the subsurface sediments. The GHPs are located within an area of 10 km2 uniformly influenced by one sediment source and ocean currents implying no evidence of any appreciable heterogeneity in clayey mineral content and composition. Macroscopic observations of sediments from different pingos are in agreement with this. It means adsorption potential of bulk sediments is uniform within the area. Therefore, if some adsorbed gas contaminated our measurements, this contamination is likely uniform throughout the whole set of the samples. Thus, the trend of lower methane concentration in GHP5 compared to other GHPs should remain. Head space

methane concentrations is one line of evidence for different methane seepage activity and geochemical conditions in pingos along with reflective seismic data, echosounder data, pore water chemistry results and video surveys. Complex interpretation of these data supports our conclusion of modest methane supply in superficial sediments of GHP5 as opposed to larger methane discharge in other GHPs.

There is at least one paper on free methane concentrations in sediments off Spitzbergen, that did not use sodium hydroxide pre-treatment for dissolved methane measurements, and should be cited Kneis et al. (2004). These authors analysed 26 sediment samples between 15 and 30 cm depth and found a thermogenic methane signature, d13C of -50.8 (mean) in the adsorbed methane but a d13C of -65.2 (mean) in the free methane. The free methane concentrations, 0.5 – 5.5 micromol/litre were lower than measurements from a similar sediment depths in the pingo areas, 6-330 micromol/litre (Serov et al. 2017). It was suggested, Kneis et al. (2004), that the adsorbed methane was not available to the microbes and that the free methane was probably a mixture from both thermogenic and biogenic sources. However, the methane isotope data reported from the pingo area (Serov et al. 2017) was taken from a greater sediment depth so that a direct comparison of the sources of the free methane cannot be made.

Response: We appreciate the suggestion to refer to a paper of our colleague from CAGE Dr. Knies. Despite, unravelling the source of gas is indeed an important topic, it is not a focus of our study. We submit that expanding the discussion chapter of our paper to cover isotopic compositions of adsorbed and dissolved gas in bottom sediments around Svalbard archipelago would dilute the focus of our work. However, one important conclusion may be drawn from comparing results of Knies et al., and Serov et al.,: concentrations of methane in pingos at the same subsurface depth are up to 660 times higher. As opposed to regional study of Knies et al., not targeting seeps, our study site demonstrates drastically different style of methane release with greater abundance of labile methane detected not only geochemically, but with direct and indirect geophysical observations.

[Figure]

Ertefai, T. F, Heuer, V B, Prieto-Mollar, X., Vogt, C., Sylva, S. P, Seewald, J. S., Hinrichs, K-U. (2010). The biogeochemistry of sorbed methane in marine sediments. Geochimica et Cosmochimica Acta, 74: 6033-6048

Knies, J., Damm, E., Gutt, J., Mann, U., Pinturier, L. (2004) Near-surface hydrocarbon anomalies in shelf sediments off Spitsbergen: Evidences for past seepages. Geophysics, Geochemistry, Geosytems 15: Q06003, doi:10.1029/2003GC000687

Serov, P., Vadakkepuliyambatta, S., Mienert, J., Patton, H., Portnov, A., Silyakova, A., Panieri, G., Carroll, M. L., Carroll, J., Andreassen, K. and Hubbard, A. (2017). Postglacial response of Arctic Ocean gas hydrates to climatic amelioration. Proceedings of the National Academy of Sciences, 114: 6215-6220.

––––––––––––––––––––––––––

---

## Referee Comment (RC3) · Anonymous Referee #2 · 25 Apr 2018

**Comment on bg-2017-540**
**Geophysical and geochemical controls on the megafaunal community of a high Arctic cold seep**
Sen A et al.

This manuscript presents data on the megafaunal community structure associated with cold seep sites in the western Barents Sea inferred from high resolution seabed imagery, and relates these data to available geochemical information. The manuscript is well-written, and as information on the ecology and biogeochemistry of cold seeps in the Arctic is still quite rare, it makes an important contribution to the field.

At the same time, there are some methodological constraints that limit the interpretability of the data. In particular, the conclusions drawn with regard to microhabitats are in my view not fully supported by the available data and the respective part of the discussion could be shortened.

Seepage rates/ volumes at cold seep sites often vary strongly over small spatial scales, and the seafloor pictures shown in this ms indicate that the same seems to be true true for the GHPs under study, with patches of bacterial mats, for example, probably indicating higher than average methane/ sulphide availability. In fact colonisation by bacterial mats and certain megafauna species (Calyptogena, Acharax; siboglinids etc), are often linked to relative flux rates (e.g. at HMMV or Hydrate Ridge) and can even serve as (crude) indicators for seepage intensity. The limited number of sediment cores taken in this study (and I fully accept the limits that can be placed on sampling in Arctic and deep waters) will likely not have been sufficient to resolve this spatial pattern which is likely to impact on microhabitat preferences.

In addition, temporal variability of seepage - even at sites considerably deeper than those under investigation here - is often strongly linked to tidal rhythms and longer term observations are likely to be necessary to conclude with certainty whether active seepage occurs at a specific site or not.

In this context I was also surprised to see that all pictures taken along each of the specific transects seem to have been 'lumped together' for analysis and no attempt was made to distinguish between fauna at more or less active seepage sites.

Judging from fig. 1, the photographic transects seem to have included (or at least could have been extended to include) reference areas without seepage and I was disappointed to see no comparison between fauna at seep and reference sites. Is there a reason for this ? This would add a very valuable and relevant dimension to the manuscript.

While it is likely that AOM consortia provide the sulphide utilised by chemoautotrophic bacteria, it seems unlikely that sulphide can be both abundant enough to support the bacterial mats visible in the photographs while being removed so efficiently from the substrate that concentrations in the upper sediment and bottom water were below detection limit. Could this be an analytical error ?

---

## Author Comment (AC3) · 27 Apr 2018

**Comment on bg-2017-540 Geophysical and geochemical controls on the megafaunal community of a high Arctic cold seep**

Sen A et al.

This manuscript presents data on the megafaunal community structure associated with cold seep sites in the western Barents Sea inferred from high resolution seabed imagery, and relates these data to available geochemical information. The manuscript is well-written, and as information on the ecology and biogeochemistry of cold seeps in the Arctic is still quite rare, it makes an important contribution to the field.

Response: Thank you for appreciating the manuscript.

At the same time, there are some methodological constraints that limit the interpretability of the data. In particular, the conclusions drawn with regard to microhabitats are in my view not fully supported by the available data and the respective part of the discussion could be shortened.

Response: We hope that we have addressed your specific concerns adequately below. Please note that we were not referring to or studying microhabitats in this manuscript. We were mainly looking at larger scale patterns and processes, as opposed to small scale variability in habitats and conditions.

Seepage rates/ volumes at cold seep sites often vary strongly over small spatial scales, and the seafloor pictures shown in this ms indicate that the same seems to be true true for the GHPs under study, with patches of bacterial mats, for example, probably indicating higher than average methane/ sulphide availability. In fact colonisation by bacterial mats and certain megafauna species (Calyptogena, Acharax; siboglinids etc), are often linked to relative flux rates (e.g. at HMMV or Hydrate Ridge) and can even serve as (crude) indicators for seepage intensity. The limited number of sediment cores taken in this study (and I fully accept the limits that can be placed on sampling in Arctic and deep waters) will likely not have been sufficient to resolve this spatial pattern which is likely to impact on microhabitat preferences.

Response: We completely agree that seepage and concentrations/fluxes of different compounds demonstrate heterogeneity over very small spatial scales at chemosynthesis based habitats such as seeps. We also agree that animal and microbial distributions can therefore serve as crude indicators of seepage patterns. However, as mentioned in our response to Dr. Dando, we did not attempt to use the images or the cores to examine patterns over small spatial scales. Our aim was to compare the different gas hydrate pingos, specifically to compare pingo 5 to the other three pingos, which are on the scale of several hundred meters in diameter. Similar numbers of cores were taken from pingo 5 as well as from the others, and we believe that these cores are representative of conditions at those pingos as a whole, which is what we examine and discuss. Furthermore, we used additional lines of evidence, in addition to geochemical measurements, to support our conclusions of inter-pingo differences, such as presence/absence of gas flares and seismic data that demonstrates different types of sub-surface gas/gas hydrate reservoirs.

An analogy can perhaps explain this issue. Mountain tops and valleys have very different kinds of vegetation because they experience different environmental conditions overall, and this difference corresponds to what we were demonstrating, i.e. that pingo 5 has a different animal community from the other pingos that corresponds with overall different geochemical conditions. For example, high altitude trees, say, pines, might be absent in the valleys, which is similar to the siboglinids being absent from pingo 5. Within a mountain top or a valley, there are additionally small scale differences in species distributions related to various factors, such as soil type and quality, the presence of streams or rivers, steepness of slope, etc. Similarly, within a pingo that contains siboglinid worms, small scale differences can determine if the worms are present in a location or absent a few centimeters away. We would like to emphasize that we are not tackling this second question and we agree that our methodology is insufficient for doing so.

In addition, temporal variability of seepage - even at sites considerably deeper than those under investigation here - is often strongly linked to tidal rhythms and longer term observations are likely to be necessary to conclude with certainty whether active seepage occurs at a specific site or not.

Response: It is true that we cannot assess temporal variability in seepage. However, as addressed in Dr. Dando's review, in terms of gas flares, we are confident that pingo 5 does not show any flare activity because this data was collected over multiple years, across different seasons, and each time, over long periods of time that well encompassed tidal rhythms. Furthermore, we do not say that pingo 5 exhibits no active seepage in the sediment, quite the contrary, in fact, we argue that pingo 5 does show enhanced flux of methane and sulfide , which could even account for the bacterial mats seen there (page 11, lines 18-19). Instead, what we argue is that methane flux, sulfate flux and sulfide flux and therefore AOM rates, are lower at pingo 5. In addition, we did not recover any gas hydrates from the +5 gravity cores at pingo5 whereas we recovered gas hydrates from the gravity cores from the other pingos. Such difference in gas hydrate abundance also clearly demonstrates the contrast in methane supply from pingo 5 compared to the other pingo features.

In this context I was also surprised to see that all pictures taken along each of the specific transects seem to have been 'lumped together' for analysis and no attempt was made to distinguish between fauna at more or less active seepage sites.

Response: In fact, this is what we did and was the aim of the manuscript: to compare the 'less active' pingo 5 to the 'more active' pingos 1-3.

Judging from fig. 1, the photographic transects seem to have included (or at least could have been extended to include) reference areas without seepage and I was disappointed to see no comparison between fauna at seep and reference sites. Is there a reason for this ? This would add a very valuable and relevant dimension to the manuscript.

Response: We did compare with images that were far enough away from the pingos (at least 1 km away) to be considered more representative of 'reference' sites. However, bacterial mats were nonetheless seen in this area and since we are unaware of the extent to which lateral diffusion takes place, we cannot confidently state that any of our image transects included true

reference, non-seep impacted areas. That is why even though we did make some comparisons to 'non-seep' areas, we were very cautious with these comparisons.

While it is likely that AOM consortia provide the sulphide utilised by chemoautotrophic bacteria, it seems unlikely that sulphide can be both abundant enough to support the bacterial mats visible in the photographs while being removed so efficiently from the substrate that concentrations in the upper sediment and bottom water were below detection limit. Could this be an analytical error ?

Response: As we clarified for Dr. Dando, the detection limit was 40 µM. Bacteria can survive on lower concentrations, therefore, they might be present even when we are unable to quantify sulfide through our measurements. As we agreed to do, based on Dr. Dando's discussion of this topic, we will remove the section that says that sulfide flux to the bottom water is so low as to allow non-seep specific fauna from colonizing the area without getting poisoned by sulfide, since it is true that sulfide might be reaching the bottom water despite our inability to detect it.

---

## Author Response (AR2)

Suggestions for revision or reasons for rejection (will be published if the paper is accepted for final publication)

This manuscript is a useful addition to the literature on the fauna at Arctic seeps, although it is regrettable that it does not include information on the background sediment and drop-stone fauna, at these depths, this would have better revealed the seep fauna characteristics. The revised manuscript is greatly improved and it is now a lot clearer how the study was organized. However, there are still some areas that need further clarification, reconsideration and and re-wording.

Response: Thank you for appreciating the changes made. We have made changes as suggested, which are outlined below.

The main finding was that Oligobrachia were absent from Pingo 5 that showed no visible gas flares and had lower methane concentrations below 50 cm sediment depth. However, since isolated tubes could not be detected it is not possible to prove the species was not present.

Response: As we mentioned in an earlier response, it is possible that a few, isolated or scattered individuals of *Oligobrachia* are present and were not detectable in the images from pingo 5. This is a known and accepted limitation of image based surveys (although as we also mentioned, we did not retrieve any *Oligobrachia* from cores from pingo 5 either). However, dense aggregations were clearly not present and that is what we focused on. We have now changed the text to say 'frenulate aggregations' as opposed to simply frenulates to make this clearer.

Supplementary Fig. 1 now explains how the frenulates were observed in the camera images. The authors still need to explain how the "mottled sediment" in this Figure was identified as frenulate tubes, since it is not clear, even on enlarging the supplied figure, that the blotches are frenulate tubes. If the tubes were apparent on enlargement of the original images then an example of this should be shown in the supplementary figure. Alternatively, one of the cores containing Oligobrachia may have been placed into this mottled seabed to allow identification and this should be described.

Response: In the Methods section, we do mention that samples were taken from among the 'blotches' or 'mottled sediment' and they were identified as being frenulate aggregations. This is pretty standard: Gebruk et al., 2003; Paull et al., 2015; Rybakova (Goroslavskaya) et al., 2013, etc. also saw similar aggregations, sampled them and using those samples determined what species constituted the aggregations and applied that identification to visually similar blotches in the sediment. We employed the same methodology and believe that it is adequate. We acknowledge that there are issues with identifying species in images, however, we are confident we identified species, including aggregations of *Oligobrachia* to the best of our abilities. Please note also, that similar identifications of *Oligobrachia* patches have been made and published from the same cruise and therefore in the same manner (Åström et al., 2016).

It is not stated whether the Supplementary Figure is from a tow-cam or a ROV camera image. Since the ROV camera was of lower resolution than the tow-cam it would be useful to have images, showing the frenulate clusters, from both cameras in the Figure, especially since there is not an explicit statement that Oligobrachia clusters could be seen in the ROV images from Pingo 3. This is needed since, unlike the pingos with active gas flares, pingo 5 was only imaged using the ROV camera and the larger Nothria polychaetes were not seen in the ROV images (P7). On P15 it is stated that "GHP5 was surveyed with the ROV before mosaic based imaging was conducted with the explicit purpose of locating siboglinid worms, since they were considered to be representative of locations with active seepage. Despite these efforts, no aggregations of these animals were seen." However,

there is still no statement that siboglinids were seen using the ROV on other pingos. It would also be useful to list the cores in which frenulates were recovered.

Response: We do indirectly mention that *Oligobrachia* was seen in the pingo 3 ROV images because we go through and list the animals not seen in these images (and *Oligobrachia* was not one of these species not seen in the pingo 3 ROV images). However, now, we clearly state that *Oligobrachia* was visible in both the ROV images and the tow cam images. To make this point clearer, we have changed the supplementary figure 1 showing *Oligobrachia* to include both an ROV image as well as a tow cam image to demonstrate that aggregations of *Oligobrachia* were visible in the ROV images from pingo 3 (raw images are provided). With respect to the cores and samples, worms were recovered both in cores used in this study as well as additional cores and scoop samples not used in this study. We now mention this, because it is important to keep in mind that more samples or data than is relevant to this specific study supports the point of abundant vs. no aggregations of *Oligobrachia* at pingo 5. Additionally, we revisited the site in 2017 with an even higher resolution tow cam system and during this cruise as well, no *Oligobrachia* worms were seen at pingo 5. We include this observation now in the discussion to further demonstrate why we truly believe that *Oligobrachia* aggregations are not present at pingo 5.

An absence of Oligobrachia from Pingo 5, if real, is difficult to explain since, as the authors state, the sulphide gradient in all the cores is similar and O. haakonmosbiensis obtains nutrition from the endosymbiotic sulphur-oxidising symbionts. The authors propose that methane "serves as a carbon source for frenulate worms. The pingo currently lacking a large sub-surface gas source and lower methane concentrations likely has lower sulfide flux rates and limited amounts of carbon, insufficient to support frenulates." There is no evidence in the literature that DIC is limiting for any frenulate and, although DIC was measured in the present study, no comparative DIC depth profiles are shown. (In any case the frenulates will produce DIC through respiration and this could lessen the demand.)

Response: We agree that explaining the absence of *Oligobrachia* aggregations is difficult and therefore we extend hypotheses based on the little information that currently exists for this taxon and the information we have about the study site. It is true that we do not know about how DIC availability affects *Oligobrachia*, therefore we simply say that it is possible that microbially mediated DIC (and/or DOC) could be a possible limiting factor given that: 1) other researchers have suggested that DIC and/or DOC could make up an important aspect of *Oligobrachia* nutrition (Lösekann et al., 2008), 2) pingo 5 has lower sediment methane concentrations, 3) pingo 5 likely has lower methane flux (seismic evidence) and 4) pingo 5 has a different microbial community with lower abundance of ANMEs.

Similarly neither sulphide flux rates nor sulphate reduction rates are presented. Sulphide flux rates/sulphate reduction rates will influence frenulate density but are unlikely to be a reason why they are absent from pingo 5, given the available sulphide data and the presence of bacterial mats on part of pingo 5, indicating active seepage. Sulphate reduction rates in the O. haakonmosbiensis zone on the HM mud volcano are extremely variable (Felden et al. 2010), suggesting that high fluxes are not necessary for the species. The rates peaked at 50-75 cm depth, consistent with the penetration depth of the frenulates there (de Beer et al. 2006).

Response: We acknowledge that certain measurements are missing from our dataset. However, we do have sulfate flux rates and we use them to get an understanding of sulfide production rates. We do not go into detail about sulfide flux rates affecting *Oligobrachia*, but simply mention that it could be a possibility, since our sulphate flux rates indicate lower sulphide fluxes at

pingo 5 and we reference articles that have shown that *Oligobrachia* has been seen associated with high sulphide flux rates. Additionally, we consider DIC/DOC availability, in relation to methane fluxes as a possible explanation for the absence of *Olgiobrachia* clusters from pingo 5. In short, we covered different angles and took alternative hypotheses into consideration because it is true that explaining the absence of *Oligobrachia* worms from pingo 5 is not an easy task.

The discussion on the possible role of DOM uptake as a requirement for the frenulates (P17-18) is interesting but ignores the most probable explanation for the large 13C depletion in O. haakonmosbiensis, i.e. that the DIC utilized by the endosymbionts is mainly derived from methane oxidation, as shown by the 13C depletion of the authigenic carbonates (Hong et al. 2017). This DIC would be further depleted by the frenulate symbionts: see the discussion in Spiro et al. (1986).

Response: In fact, we do discuss DIC uptake as a possible explanation for 13C depletion in *O. haakonmosbiensis* and from that, we discuss that DIC could be a limiting factor explaining the absence of *Oligobrachia* from pingo 5 (please see paragraph 2 on page 17 and also see page 18).

Both Nothria and sipunculids were absent from Pingo 5, was this due to lack of resolution of the ROV images or to some other cause?

Response: This is actually explained in the Methods section. Briefly, the absence of Nothria from pingo 5 was likely a resolution issue, but this is not the case for sipunculids since they were visible in the pingo 3 ROV images.

The authors provide a large number of references to justify their use of the term megafauna (i.e. very large fauna), in place of macrofauna. It is equally possible to provide references for the use of macrofauna for the range of species described, including a recent paper by three of the co-authors who refer to similar fauna as "macrobenthos" (Åström et al. 2016). In the present manuscript megafauna is defined as "animals large enough to be seen with the naked eye", this definition would include the larger meiofauna (commonly used to refer to animals passing through a 0.5 mm sieve) and therefore, apparently, leaving no animals in the macrofauna. On P19 the authors state that "Details on the macrofaunal community composition is currently being compiled" and mention thyasirid bivalves as having being found, these are certainly "visible to the naked eye". How do the authors therefore define "macrofauna" since there does not appear to be a consistency in their terminology? Later in the manuscript Amon et al. (2017) is cited that the smallest megafauna should have a minimum dimension of 1 cm.

Response: As mentioned in an earlier response as part of this review process, no clear distinction exists with respect to the terms macrofauna and megafauna. Indeed, definitions of macrofauna include set lower limits, but no upper limit to animal sizes is defined. Therefore, there can and often is, overlap between the two categories. We simply wanted to demonstrate that our use of the term megafauna is not incorrect and in fact, consistent with a number of other studies. It should be noted that in the Åström et al., 2017 paper, macrofauna was used to refer to animals in sediment samples that were retained on a 0.55 mm sieve and megafauna was used to refer to animals visible in images. Since this point can be argued back and forth, we have now changed the text in the manuscript such that we state early on that in this manuscript, we refer to the fauna in this study, which are visible in photographic images, as megafauna (which is consistent with a number of other studies). In this way, we acknowledge that terminology preferences can differ on a

person by person basis, but we choose to use one term ourselves, and our choice aligns with a number of other studies using similar methods.

P3 "this study is particularly useful for teasing apart the factors affecting the large-scale distribution of chemosynthesis based species, since these animals are directly reliant on seeping chemicals." On continental shelves and slopes chemosynthesis-based species are frequently found in non-seep reducing sediments, indeed most of the frenulate species are not found at seeps, the reducing chemicals coming from the breakdown of organic matter in the upper sediment. The authors show that they appreciate this in the Discussion on P13. However, we still do not know whether O. haakonmosbiensis is a seep obligate species or also occurs in lower densities in reducing sediments.

Response: True, we do not know if *Oligobrachia* is seep obligate or not and we mention this in the discussion. Determining this is however, beyond the scope of this study, but we acknowledge that it is an important open question.

P4 "no such free gas emissions were seen over GHP5 during repeat on-site observations over 3 years (2013-2016) across different seasons." Since the observations were not continuous it would be useful to add the amount of time spent on-site at pingo 5.

Response: We do not have an ability to show amount of time spent over particular locations because the precise timing was not recorded. In figure 1 we show the areas covered with mosaics and the locations of gravity cores. More gravity cores were taken at GHP5 than at each of the other pingos, plus we collected images for three mosaics in additional to general surveying efforts at GHP5. Therefore we did not spend less time acquiring data from GHP5 compared to the other pingos. Multibeam and single beam echosounder data were recorded continuously, therefore, we are confident in the validity of our observations of no free gas ebullition at GHP5. Moreover, papers cited in this work (Serov et al., 2017, Hong et al., 2017, Hong et al., 2018) include observations from other research cruises conducted at different years and over different seasons reporting the same pattern. In 2017 we acquired a 3D seismic cube over the entire area shown in Figure 1a with 50 m spacing between the survey lines and constant sailing speed of 4knots (Waage et al., in prep.). This means that every pingo was within the multibeam echosunder swath at least 10-12 times, depending on the pingo size. The echosounder data from this survey showed exactly the same distribution of flares as we report in our manuscript.

P8 L10 "The detection limit for this method is 40 µM." This should read: " The detection limit for the variant of the method used was 40 µM", since the original method (Cline, 1969) allows detection of sulphide as low as 1.0 µM. A limit of 0.1µM is achievable using reduced reagent amounts.

Response: We have changed this accordingly.

Figure 4a: the colours for GHP5 920 and GHP5 1048 are not easily distinguished. Methane concentration – is the volume sediment or pore water, since whole sediment samples were taken?

Response: Agreed, pink and red are close. We have changed the colours now. Sediment for methane headspace measurements were taken at intervals along the length of each core. We collected 5ml of sediment at each interval. This is now clarified in the Methods text.

Figure 5: change the symbols to be consistent with Fig. 4

Response: We have done this now.

de Beer, D., Sauter, E. J., Niemann, H., Kaul, N., Foucher, J.-P., Witte, U., et al. (2006). In situ fluxes and zonation of microbial activity in surface sediments of the Håkon Mosby Mud Volcano. Limnology and Oceanography, 51(3), 1315-1331.

Felden, J., Wenzhöfer, F., T., F., & Boetius, A. (2010). Transport and consumption of oxygen and methane in different habitats of the Håkon Mosby Mud Volcano (HMMV). Limnology and Oceanography, 22, 2366-2380.

Spiro, B., Greenwood, P. B., Southward, A. J., & Dando, P. R. (1986). 13C/12C ratios in marine invertebrates from reducing sediments: confirmation of nutritional importance of chemoautotrophic endosymbiotic bacteria. Mar. Ecol. Prog. Ser., 28, 233-240.

[revised manuscript text omitted]

▲ free gas emitting GHPs  ▼ GHP5

[Figure]

[Figure]

[Figure]

[Figure]

[Figure]

[Figure]

[Figure]

(a)

20 cm

[Figure]

(b)

20 cm